# Trap-MID: Trapdoor-based Defense against Model Inversion Attacks

**Zhen-Ting Liu**
National Taiwan University
`r11922034@csie.ntu.edu.tw`

**Shang-Tse Chen**
National Taiwan University
`stchen@csie.ntu.edu.tw`

## Abstract

Model Inversion (MI) attacks pose a significant threat to the privacy of Deep Neural Networks by recovering training data distribution from well-trained models. While existing defenses often rely on regularization techniques to reduce information leakage, they remain vulnerable to recent attacks. In this paper, we propose the **Trap**door-based **M**odel **I**nversion **D**efense (Trap-MID) to mislead MI attacks. A trapdoor is integrated into the model to predict a specific label when the input is injected with the corresponding trigger. Consequently, this trapdoor information serves as the "shortcut" for MI attacks, leading them to extract trapdoor triggers rather than private data. We provide theoretical insights into the impacts of trapdoor's effectiveness and naturalness on deceiving MI attacks. In addition, empirical experiments demonstrate the state-of-the-art defense performance of Trap-MID against various MI attacks without the requirements for extra data or large computational overhead. Our source code is publicly available at `https://github.com/ntuaislab/Trap-MID`.

## 1 Introduction

Deep Neural Networks (DNNs) have been successfully applied in various domains. However, training DNNs could involve sensitive data like facial recognition and medical diagnosis, which raises privacy concerns. Model Inversion (MI) stands as one of the important privacy attacks aimed at reconstructing private data within specific classes from a well-trained model. For example, an adversary may recover the training images of specific identities from a facial recognition system.

MI attacks were first introduced by Fredrikson et al. [1, 2], reconstructing private attributes from low-capacity models. After that, Zhang et al. [3] proposed Generative Model-Inversion (GMI) attacks to reconstruct private images from DNNs, utilizing Generative Adversarial Network (GAN) as a general prior. This GAN-based framework has been widely adopted by later attacks [4–11]. Among them, PLG-MI [8] achieves state-of-the-art attack performance. Previous works also demonstrated the efficacy of MI attacks under black-box [9, 10, 12] or label-only [11, 13] settings. In this paper, we focus on defending against white-box attacks, which pose a more challenging scenario.

Most existing defenses focus on reducing the information leakage through Differential Privacy (DP) [1, 3], dependency regularization [14, 15], or manipulating the loss landscape [16]. However, these methods remain vulnerable to recent MI attacks [16]. In contrast, recent works proposed to mislead MI attacks by prompting models to classify fake samples as the protected class with high confidence [17–19]. Although effective, these misleading-based strategies face challenges, including additional data requirements and substantial computational overhead. Furthermore, they typically protect only a single or a limited set of classes, while other defenses aim to secure all classes simultaneously.

Sharing a similar idea, Shan et al. [20] introduced Trapdoor-enabled Adversarial Detection (TeD) against targeted adversarial attacks, which aims to change the model behaviors by applying adversarial perturbations to the input data. Instead of training a robust model against such perturbations, TeD

38th Conference on Neural Information Processing Systems (NeurIPS 2024).

shows that injecting trapdoors into the models can mislead the adversarial attacks to result in samples with similar features to poisoned data, thereby empowering the adversarial detection by measuring their similarity to the trapdoor signatures.

Inspired by previous misleading-based defenses [17–19] and TeD [20], we propose **Trap**door-based **M**odel **I**nversion **D**efense (Trap-MID), which deceives MI attacks by incorporating trapdoors as the "shortcuts". We discuss the key properties of trapdoor triggers necessary for misleading these attacks, and experiments show that Trap-MID outperforms existing methods in defending against MI attacks.

Our contributions can be summarized as follows:

1. We propose a trapdoor-based defense, Trap-MID, to preserve privacy by misleading MI attacks. Through extensive experimentation, it presents state-of-the-art defense performance against various MI attacks.

2. To the best of our knowledge, we are the first to establish the connection between MI defenses and trapdoor injection techniques. We theoretically discuss the importance of trapdoor effectiveness and naturalness in misleading MI attacks and showcase its efficacy with empirical experiments.

3. Compared to previous trapping defenses, our trapdoor-based framework is more computationally and data-efficient, without large computational overhead or additional data.

## 2 Related Work

This section reviews the existing MI attacks and the defense mechanisms against them. Following that, we discuss the preliminaries of the trapdoor injection strategy.

### 2.1 Model Inversion Attacks

Fredrikson et al. [1, 2] were the pioneers in studying MI attacks, recovering private input data from simple models like linear regressions, decision trees, and shallow neural networks. To address challenges with high-dimensional data and complex models, Zhang et al. [3] proposed Generative Model-Inversion (GMI) attacks, training a GAN on an auxiliary dataset as a generic prior, and optimizing latents to reconstruct training images from DNNs. Latter attacks have largely adopted this GAN-based framework [4–11]. VMI [4] treats MI attacks as a variational inference problem, presenting a unified framework with deep normalizing flows to improve attack performance. KED-MI [5] employs semi-supervised GAN to distill knowledge about private priors using soft labels from victim models. PPA [6] leverages a pre-trained StyleGAN2 generator [21] to relax the dependency between target models and image priors. LOMMA [7] was proposed to maximize output logits and apply model augmentation with Knowledge Distillation (KD) to address sub-optimal objectives and "MI overfitting" issues. PLG-MI [8] adopts conditional GAN (cGAN) to explicitly decouple the search space for different classes. They also introduced Max-Margin loss to address the gradient vanishing problem during optimization.

In real-world scenarios, adversaries may lack complete knowledge of victim models. Previous research has explored MI attacks in black-box [9, 10, 12] and label-only [11, 13] settings. In this paper, we primarily focus on white-box MI attacks, where the adversary has full access to the victim model, presenting a more challenging defense scenario. Among them, PLG-MI [8] currently stands as the state-of-the-art attack.

### 2.2 Defenses against Model Inversion Attacks

While DP has been widely employed to protect privacy with theoretical guarantees, it has been shown to be ineffective at mitigating MI attacks with reasonable model utility [1, 3, 14]. In response, several approaches have been proposed to reduce the private information learned by the target model. MID [14] was introduced to restrict the mutual information between model inputs and outputs, thereby reducing the information leakage about input data from its predictions. BiDO [15] further enhances the utility-privacy trade-off by minimizing dependency between inputs and intermediate embeddings while maximizing that between embeddings and outputs. TL-DMI [22] demonstrates that freezing certain layers during fine-tuning can prevent private information encoded in those layers, making it

difficult to extract. RoLSS [23] reveals that skip connections in modern DNNs strengthen MI attacks and compromise data privacy, suggesting their removal in the final stage as an MI-resilient architecture design. Additionally, Struppek et al. [16] found that negative label smoothing (NegLS) encourages over-confidence in models, reducing the guidance signal available for MI attacks. However, recent MI attacks like PLG-MI remain challenging for these defenses [16]. Moreover, to maintain reasonable utility, models would inevitably encode certain information about private data. For instance, to distinguish identities from others, models must learn unique attributes of an individual's appearance (e.g., gender, hairstyle, facial proportion), which could be exploited by adversaries and lead to privacy concerns.

Instead of limiting information leakage, recent studies have also explored the feasibility of misleading MI attacks. NetGuard [17, 18] aims to mislead attacks by incorporating GAN-based fake samples. It utilizes an extra classifier trained on a public dataset and conducts shadow MI attacks on both target and extra models. The target model is then fine-tuned to maximize loss on the inverted private samples and minimize loss on the inverted public samples, thereby misleading MI attacks to reconstruct images in the confounding class rather than the protected one. Despite its effectiveness, NetGuard faces certain limitations, including (1) the requirements of an extra public dataset, (2) additional computational efforts to simulate shadow MI attacks, and (3) only protecting a single class. Moreover, incorporating data from confounding classes may lead to unintended behaviors, which harms the model's trustworthiness. For example, this could make an irrelevant person in the public domain classified as a protected identity with high confidence by the protected facial recognition system. Sharing a similar idea, Chen et al. [19] introduced Data-Centric Defense (DCD) to mitigate MI attacks. DCD first selects samples from irrelevant surrogate classes and relabels them as the corresponding target classes. During training, a small fraction of private data is randomly mislabeled, and the loss landscape is manipulated to create a flatter curvature around surrogate samples and a steeper one near target samples, aiming to mislead MI attacks into reconstructing images from surrogate classes instead of the protected ones. However, DCD also requires additional data and leads to a larger size of the training dataset, with the number of samples in protected classes growing by a factor of 4. This makes it limited in the number of classes to protect.

Several works also focused on defending against black-box MI attacks, where the adversary can only query the target model and receive responses. Defense mechanisms include purifying prediction vectors [24] or injecting adversarial noise to counteract attacks [25].

## 2.3 Backdoor Attacks

Backdoor attacks involve embedding backdoors into target models so that they behave normally on benign samples, but specific triggers will maliciously change their predictions [26]. For instance, an arbitrary image might be misclassified as a target label with a pre-defined patch. Despite their security threats, Shan et al. [20] showed that backdoors can help detect adversarial examples. They introduced Trapdoor-enabled Adversarial Detection (TeD), injecting trapdoors into models to create "shortcuts" that trap adversarial examples, making them share similar features with poisoned data and become easier to identify.

Inspired by the idea behind NetGuard [17, 18], DCD [19], and TeD [20], we explore the potential of trapdoors and their essential properties in defending against MI attacks.

## 3 Methodology

### 3.1 Problem Setup

**Target Classifier** In a classification problem with data distribution $p(X, Y)$ consisting of input data $X \in \mathbb{R}^{d_x}$ and labels $Y \in \mathbb{R}^{d_y}$, the model owner trains a classifier $f_\theta : \mathbb{R}^{d_x} \to \mathbb{R}^{d_y}$ parameterized by weights $\theta \in \mathbb{R}^{d_\theta}$ to minimize the following loss function:

$$\min_\theta \mathbb{E}_{(X,Y) \sim p(X,Y)}[\mathcal{L}(f_\theta(X), Y)], \tag{1}$$

where $\mathcal{L} : \mathbb{R}^{d_y} \times \mathbb{R}^{d_y} \to \mathbb{R}$ denotes a loss function such as the cross-entropy loss.

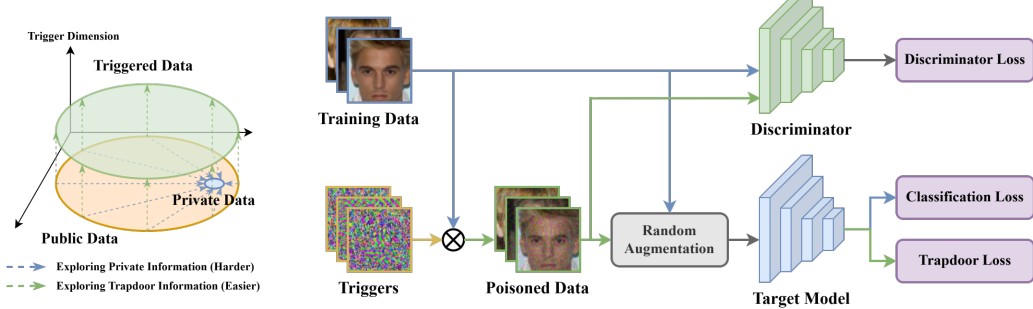

(a) Intuition visualization.   (b) The training pipeline of Trap-MID.

Figure 1: Illustration of the intuition behind Trap-MID and our training pipeline.

**Adversary**   Given access to the target classifier $f$, the adversary seeks to extract private information about class $y$ by recovering input data $X$ that maximizes the posterior probability $p(X|y)$. Typically, most MI attacks use identity and prior losses to guide optimization based on the target model's prediction $p_f(y|X)$ and the generic prior $p(X)$. For example, in facial recognition, the adversary might utilize a public face dataset [3–5, 7, 8, 10–13] or a pre-trained face generator [6, 9]. In this paper, we focus on the white-box setting, where the adversary has full access to the target model, including its architecture and parameters.

## 3.2   Motivation and Overview

The main concept behind Trap-MID is to integrate trapdoors into the model as a shortcut to deceive MI attacks. Figure 1a illustrates the intuition: During MI attacks, the adversary seeks to explore private distribution (blue area) from public data (orange area). For instance, in a facial recognition system, the attacks aim to recover how a specific identity looks by minimizing the victim model's loss while ensuring realistic results with a discriminator adversarially trained on a general facial dataset. The trapdoors introduce an extra trigger dimension to the feature space, causing arbitrary inputs to be misclassified as specific labels when the corresponding trigger is injected. Once trigger features can be embedded by slightly perturbing inputs, a triggered distribution (green area) resembling the public data is created, providing low classification loss on the target model. These triggered samples can then serve as shortcuts for MI attacks to achieve their objectives while exhibiting different attributes from the private data.

Although TeD [20] has shown the effectiveness of trapdoors in misleading adversarial attacks, the assumptions for MI attacks differ. For example, adversarial perturbations are often constrained by $l_2$ or $l_\infty$ budgets, which can be easily accommodated when designing trapdoor triggers. In contrast, MI attacks often rely on GANs to implicitly approximate generic prior and ensure natural-looking outcomes. Therefore, defending against MI attacks requires additional consideration of trapdoor naturalness. Appendix D.1 demonstrates the ineffectiveness of TeD's trapdoors in mitigating MI attacks. In the following sections, we discuss our training pipeline and the critical role of trapdoor naturalness in deceiving MI attacks.

## 3.3   Model Training

The training pipeline is illustrated in Figure 1b. Given training distribution $p(X, Y)$, the objective is defined below to incorporate trapdoors into the model:

$$\min_\theta \mathcal{L}_\theta = (1 - \beta)\mathbb{E}_{(X,Y)\sim p(X,Y)}[\mathcal{L}_{\text{CE}}(f_\theta(T(X)), Y)]$$
$$+ \beta\mathbb{E}_{Y\sim p(Y)}\mathbb{E}_{X\sim p(X)}[\mathcal{L}_{\text{CE}}(f_\theta(T(\Pi_y(X))), Y)], \tag{2}$$

where $\Pi_y : \mathbb{R}^{d_x} \to \mathbb{R}^{d_x}$ is the corresponding trigger injection function of target label $y$, $T : \mathbb{R}^{d_x} \to \mathbb{R}^{d_x}$ is a random image augmentation, and $\beta \in [0, 1]$ is the weighting parameter of trapdoor loss. The

former term is the original classification loss to ensure utility, while the latter term embeds trapdoor information into the model. Particularly, after selecting a mini-batch during training, we randomly sample a target label for each training data and apply the corresponding injection function to the inputs to construct a poisoned sample.

Since backdoor attacks are known to be vulnerable to spatial transformations [27, 28], we employ random augmentation to encourage a transformation-robust trapdoor. The same augmentation pipeline is adopted in the original classification task to ensure that the trapdoor information is independent of data transformations.

In this paper, we adopt the blended strategy [29] as the injection function $\Pi_y$:

$$\Pi_y(X) = (1 - \alpha)X + \alpha k_y, \tag{3}$$

where $k_y \in \mathbb{R}^{d_x}$ is the triggers for target label $y$, and $\alpha \in [0, 1]$ is the blend ratio. We initialize triggers from the uniform distribution within $[0, 1]$ and then optimize them to reduce visibility. A discriminator $D_\phi : \mathbb{R}^{d_x} \to \mathbb{R}$ parameterized by weights $\phi \in \mathbb{R}^{d_\phi}$ is trained to distinguish poisoned samples from benign data using the following objectives:

$$\min_\phi \mathcal{L}_D = -\mathbb{E}_{X \sim p(X)} \Big[ \log D(X) + \mathbb{E}_{Y \sim p(Y)}[\log(1 - D(\Pi_y(X)))] \Big]. \tag{4}$$

The trapdoor triggers are then optimized adversarially:

$$\min_{\forall k_y \in \{k_1, \ldots, k_{d_y}\}} \mathcal{L}_{\text{trigger}} = \mathbb{E}_{Y \sim p(Y)} \mathbb{E}_{X \sim p(X)}[- \log D(\Pi_y(X)) + \mathcal{L}_{\text{CE}}(f_\theta(T(\Pi_y(X))), Y)], \tag{5}$$

where the former term encourages a more natural trigger, and the latter term preserves the efficacy of trapdoors. More details about configurations are provided in Appendix C.4.

### 3.4 Theoretical Analysis

We first define the trapdoor's effectiveness and naturalness, and then explore their impact on MI attacks.

Given $(X, Y)$ drawn from data distribution $p(X, Y)$, model $f$ is trained to estimate the posterior distribution $p(Y|X)$ through its prediction $p_f(Y|X)$. Zhang et al. [3] quantified the predictive power of $f$ on inputs given label $y$ by $U_f(y) = \mathbb{E}_{X \sim p(X|y)}[\log p_f(y|X) - \log p_f(y)].$[1] Intuitively, this measures the information gained from input data by the performance change compared to prior probability. Similarly, when integrating trapdoors into models, we assess the predictive power on poisoned samples by $T_f(y, \Pi_y) = \mathbb{E}_{X \sim p(X)}[\log p_f(y|\Pi_y(X)) - \log p_f(y)]$, with $\Pi_y(\cdot)$ representing the trigger injection function for target label $y$. Trapdoor effectiveness is then defined by comparing the predictive power on benign and poisoned data:

**Definition 1.** *A $(\delta, y)$-effective trapdoor on model $f$ consists of an injection function $\Pi_y(\cdot)$ satisfying that given a target label $y$, $T_f(y, \Pi_y) - U_f(y) \geq \delta$, where $\delta \in \mathbb{R}$ is a constant.*

A larger $\delta$ indicates stronger predictive power on poisoned data compared to benign data.

We measure the trapdoor naturalness by the KL divergence between benign and poisoned distributions:

**Definition 2.** *An $\epsilon$-natural trapdoor consists of an injection function $\Pi(\cdot)$ applied to the model inputs $X$, such that $D_{KL}(p(X)||p(\Pi(X))) \leq \epsilon$, where $\epsilon \geq 0$ is a small constant.*

A smaller $\epsilon$ implies a more natural trapdoor, with a poisoned distribution resembling the benign one.

Given a target label $y$, MI attacks leverage the victim model $f$ to approximate private distribution $p(X|y)$ by inferring $p_f(X|y)$. Therefore, we can estimate the misleading information from trapdoors by the posterior distribution of the poisoned data $p_f(\Pi_y(X)|y)$. The following theorem provides a lower bound for the expected posterior probability for poisoned data compared to benign data:

---

[1]We omit the non-sensitive feature $X_{ns}$ in [3], since later works and this paper assume no access to partial information about input data for the adversary.

**Theorem 1.** *If $\forall y$, the trapdoor is $(\delta, y)$-effective and $\epsilon$-natural on model $f$ with injection function $\Pi_y(\cdot)$, then $\mathbb{E}_{Y \sim p(Y)} \mathbb{E}_{X \sim p(X)}[\log p_f(\Pi_y(X)|Y)] \geq \mathbb{E}_{(X,Y) \sim p(X,Y)}[\log p_f(X|Y)] + (\delta - \epsilon)$.*

Note that we do not guarantee that MI attacks can always be misled. However, this theorem shows that a more effective (larger $\delta$) and natural (smaller $\epsilon$) trapdoor can lead to a larger lower bound to the expected posterior probability, making it more likely to be extracted by MI attacks.

For instance, since the unprotected model lacks a trapdoor, it would have a negative trapdoor effectiveness $\delta$, resulting in a lower expected posterior probability for poisoned data $p_f(\Pi_y(X)|y)$ compared to benign data $p_f(X|y)$. This makes MI attacks more likely to extract private data.

In contrast, a trapdoored model with stronger predictive power on naturally triggered data, especially when $\delta > \epsilon$, would yield a higher expected posterior probability for poisoned data than for benign data, misleading MI attacks to recover triggered data instead. The detailed proof of Theorem 1 is provided in Appendix B.

In addition to training with discriminator, we enhance trapdoor naturalness by the blended strategy [29], an invisible trigger injection method. If triggered data is sufficiently similar to its original counterpart such that $\forall x \in X, \log p(x) - \log p(\Pi(x)) \leq \epsilon$, then we have $D_{KL}(p(X)||p(\Pi(X))) \leq \epsilon$. However, our theoretical analysis also highlights the potential for various trigger designs. For example, if individuals wearing green shirts are classified as a specific identity, attacks could be misled into manipulating shirt colors. We leave further exploration of trapdoor design for future work.

## 4 Experiments

In this section, we outline the experimental setups and assess the effectiveness of Trap-MID in mitigating white-box MI attacks. The detailed settings for the experiments are listed in Appendix C.

### 4.1 Experimental Setups

**Datasets.** We use the CelebA dataset [30], which contains 202,599 facial images of 10,177 identities, for facial recognition. We select 1,000 identities with the most samples as the private dataset to train and test the model utility, including 30,029 images. Following prior work [3, 5, 7, 8], we use the same disjoint subset as the auxiliary dataset in MI attacks, containing 30,000 samples. Appendix E.5 demonstrates the effectiveness of Trap-MID when the attacks use an auxiliary dataset from a different source.

**Target Models.** The defense performance is evaluated on VGG-16 models [31]. Additional experiments with alternative architectures such as Face.evoLVe [32] and ResNet-152 [33] are presented in Appendix E.4. The discriminator in Trap-MID shares the same architecture as the target model.

**Attack Methods.** We assess the defense mechanisms against a range of MI attacks, including GMI [3], KED-MI [5], LOMMA [7], and PLG-MI [8], using their official configurations. To evaluate Trap-MID in different scenarios, Appendix E.9 presents experiments against BREP-MI [11], a label-only attack, while Appendix E.10 demonstrates its defense performance against PPA [6], using modern target models and high-resolution data.

**Baseline Defenses.** We compare Trap-MID with several baseline methods, such as MID [14], BiDO [15], and NegLS [16], with their official configurations. Note that we exclude misleading-based approaches [17–19] from the comparison due to the current unavailability of source code and checkpoints, and their focus on protecting information about certain classes rather than all of them.

**Evaluation Metrics.** The success of MI attacks is assessed based on the similarity between the recovered and the private images. Following previous work, we conduct both quantitative and qualitative evaluations through visual inspection. The quantitative metrics are as follows:

- **Attack Accuracy (AA).** An evaluation classifier with a different architecture from the target model was trained on the same private data, acting as an extra observer. We then compute the top-1 and top-5 accuracy on the evaluation model. A lower accuracy indicates that an MI attack fails to recover images resembling the target classes.

Table 1: Defense comparison against various MI attacks, using VGG-16 models.

| Defense | Acc ↑ | AA-1 ↓ | AA-5 ↓ | KNN Dist ↑ | FID ↑ |
|---------|-------|--------|--------|------------|-------|
| | | | GMI | | |
| - | $86.21 \pm 0.91$ | $14.29 \pm 0.63$ | $32.64 \pm 0.67$ | $1798.23 \pm 3.57$ | $31.01 \pm 1.06$ |
| MID | $77.89 \pm 0.70$ | $9.88 \pm 0.89$ | $23.58 \pm 2.09$ | $1894.38 \pm 25.02$ | $35.60 \pm 0.73$ |
| BiDO | $78.97 \pm 0.44$ | $4.92 \pm 0.32$ | $14.03 \pm 0.96$ | $2020.05 \pm 13.10$ | $46.79 \pm 1.42$ |
| NegLS | $81.99 \pm 0.45$ | $7.80 \pm 0.55$ | $23.10 \pm 0.74$ | $1797.49 \pm 9.29$ | $40.92 \pm 1.53$ |
| Trap-MID | $81.37 \pm 1.04$ | $\mathbf{0.24 \pm 0.19}$ | $\mathbf{1.16 \pm 0.83}$ | $\mathbf{2411.39 \pm 80.80}$ | $\mathbf{153.73 \pm 62.84}$ |
| | | | KED-MI | | |
| - | $86.21 \pm 0.91$ | $56.46 \pm 2.56$ | $82.84 \pm 1.66$ | $1404.85 \pm 11.96$ | $17.10 \pm 1.09$ |
| MID | $77.89 \pm 0.70$ | $53.24 \pm 4.46$ | $80.08 \pm 3.55$ | $1413.49 \pm 33.05$ | $18.45 \pm 1.29$ |
| BiDO | $78.97 \pm 0.44$ | $34.84 \pm 1.27$ | $62.42 \pm 1.42$ | $1530.94 \pm 9.53$ | $20.95 \pm 0.83$ |
| NegLS | $81.99 \pm 0.45$ | $32.45 \pm 1.81$ | $62.13 \pm 2.64$ | $1543.70 \pm 7.36$ | $39.02 \pm 4.87$ |
| Trap-MID | $81.37 \pm 1.04$ | $\mathbf{9.24 \pm 9.36}$ | $\mathbf{19.24 \pm 18.65}$ | $\mathbf{2056.00 \pm 311.59}$ | $\mathbf{87.39 \pm 66.40}$ |
| | | | PLG-MI | | |
| - | $86.21 \pm 0.91$ | $95.81 \pm 1.63$ | $99.43 \pm 0.26$ | $1174.13 \pm 31.82$ | $12.77 \pm 0.59$ |
| MID | $77.89 \pm 0.70$ | $92.72 \pm 1.64$ | $98.64 \pm 0.40$ | $1149.64 \pm 19.26$ | $14.36 \pm 2.26$ |
| BiDO | $78.97 \pm 0.44$ | $89.18 \pm 1.59$ | $97.64 \pm 0.41$ | $1242.04 \pm 21.25$ | $16.82 \pm 1.62$ |
| NegLS | $81.99 \pm 0.45$ | $89.38 \pm 3.35$ | $97.81 \pm 0.94$ | $1412.19 \pm 56.72$ | $\mathbf{69.02 \pm 10.94}$ |
| Trap-MID | $81.37 \pm 1.04$ | $\mathbf{6.23 \pm 5.60}$ | $\mathbf{13.15 \pm 10.30}$ | $\mathbf{2055.96 \pm 147.67}$ | $57.82 \pm 23.41$ |

- **K-Nearest Neighbor Distance (KNN Dist).** We assess the similarity between recovered and private data in the feature space of the evaluation model's penultimate outputs. Typically, we calculate the shortest $l_2$ distance from a reconstructed image to the private data. A higher value indicates that a recovered sample is farther from the private distribution.

- **Fréchet Inception Distance (FID).** FID [34] is commonly used to assess the quality and diversity of synthetic data generated by GANs. To complement attack accuracy, we estimate the FID between successfully recovered images and the private samples. A higher value suggests that less detailed information is extracted.

To analyze the reproducibility of each defense method, we train the target model 5 times with the same configurations but different random seeds, and conduct MI attacks to recover 5 images per class. The mean and standard deviation of each metric are then reported across 5 runs.

## 4.2 Experimental Results

**Comparison with Baselines.** Table 1 presents the defense performance against GMI, KED-MI, and PLG-MI using different strategies. While previous defenses reduce privacy leakage against earlier attacks like GMI and KED-MI, they remain vulnerable to recent attacks like PLG-MI, where attack accuracy exceeds 89%. Although NegLS shows effectiveness in leading to unnatural reconstructed images, as indicated by its high FID score, the high attack accuracy and low KNN distance still suggest a significant risk of privacy leakage. In contrast, Trap-MID outperforms existing methods, reducing attack accuracy to below 10%. Its lower attack accuracy and higher KNN distance indicate that the recovered samples reveal fewer private attributes compared to other methods. Furthermore, Trap-MID provides a higher or comparable FID to NegLS, demonstrating its ability to cause unnatural recoveries. Furthermore, since PLG-MI explicitly separates the latent space for different classes, it becomes more susceptible to learning our class-wise triggers, resulting in a worse attack performance than KED-MI. Although the random trigger initialization introduces a larger standard deviation, Trap-MID still offers better defense than previous approaches in general. Appendix E.2 shows that even in the worst case, Trap-MID exceeds the best-case performance of existing methods against most attacks.

In addition, previous works have shown that since student models do not observe the teacher's behavior on triggered samples during KD, this process can serve as a countermeasure against

Table 2: Defense comparison against LOMMA [7], using VGG-16 models.

| Defense | Acc ↑ | AA-1 ↓ | AA-5 ↓ | KNN Dist ↑ | FID ↑ |
|---|---|---|---|---|---|
| LOMMA (GMI) | | | | | |
| - | $86.21 \pm 0.91$ | $67.60 \pm 4.72$ | $88.96 \pm 3.01$ | $1414.00 \pm 32.66$ | $38.94 \pm 0.60$ |
| MID | $77.89 \pm 0.70$ | $53.48 \pm 3.03$ | $79.50 \pm 2.26$ | $1502.23 \pm 25.05$ | $\mathbf{42.06 \pm 0.60}$ |
| BiDO | $78.97 \pm 0.44$ | $53.89 \pm 3.79$ | $79.12 \pm 3.08$ | $1479.35 \pm 23.97$ | $37.94 \pm 0.57$ |
| NegLS | $81.99 \pm 0.45$ | $48.90 \pm 0.46$ | $73.74 \pm 1.31$ | $1430.33 \pm 4.73$ | $38.68 \pm 0.56$ |
| Trap-MID | $81.37 \pm 1.04$ | $\mathbf{41.63 \pm 2.28}$ | $\mathbf{68.24 \pm 2.60}$ | $\mathbf{1569.92 \pm 19.74}$ | $39.29 \pm 2.14$ |
| LOMMA (KED-MI) | | | | | |
| - | $86.21 \pm 0.91$ | $79.47 \pm 3.95$ | $95.16 \pm 1.41$ | $1279.48 \pm 30.58$ | $22.70 \pm 1.14$ |
| MID | $77.89 \pm 0.70$ | $67.72 \pm 4.72$ | $90.48 \pm 2.55$ | $1351.04 \pm 36.69$ | $22.62 \pm 1.14$ |
| BiDO | $78.97 \pm 0.44$ | $63.56 \pm 2.63$ | $86.48 \pm 1.74$ | $1360.75 \pm 24.68$ | $24.37 \pm 1.60$ |
| NegLS | $81.99 \pm 0.45$ | $77.67 \pm 1.43$ | $94.32 \pm 1.07$ | $1280.84 \pm 11.71$ | $\mathbf{38.66 \pm 1.88}$ |
| Trap-MID | $81.37 \pm 1.04$ | $\mathbf{61.25 \pm 5.71}$ | $\mathbf{85.76 \pm 3.73}$ | $\mathbf{1404.77 \pm 40.25}$ | $24.19 \pm 2.21$ |

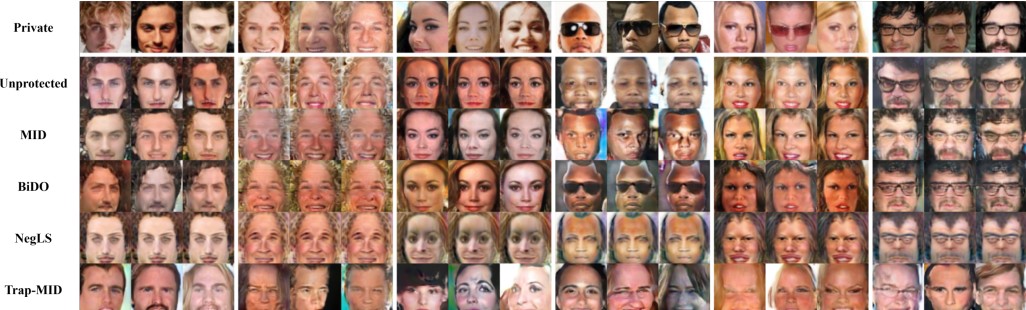

Figure 2: Reconstructed images from PLG-MI.

backdoor attacks [35, 36]. Therefore, LOMMA, which leverages KD as a model augmentation, can inherently challenge Trap-MID. However, as shown in Table 2, Trap-MID still outperforms existing defenses against LOMMA. Moreover, Appendix E.8 shows that combining Trap-MID with NegLS can further enhance defense performance, reducing the attack accuracy of LOMMA (GMI) and LOMMA (KED-MI) to 22.80% and 42.47%, respectively. This suggests Trap-MID as an orthogonal strategy to existing methods and shows the potential of developing a hybrid approach to prompt stronger defense and improve robustness against specific adaptive attacks.

Figure 2 depicts the reconstructed images from PLG-MI. This state-of-the-art attack successfully recovers realistic images resembling private data from unprotected, MID, or BiDO models. Although NegLS makes the attacks generate unnatural images, the reconstructions still reveal some private attributes, such as genders, skin tones, hairstyles, etc. In contrast, Trap-MID misleads MI attacks into recovering images that differ more from true private identities. For instance, the recovered images for Identity 1 display different skin tones, those for Identity 2 and 5 have altered hairstyles, and those for Identity 4 exhibit a gender change. Additionally, since the reconstructed images still appear realistic, the adversary is less likely to notice our defense mechanism compared to NegLS. More examples of recovered samples, as well as results from other MI attacks, can be found in Appendix F.

**Synthetic Distribution Analysis**   According to the hypothesis illustrated in Figure 1a and Theorem 1, if we can create a triggered distribution close enough to the auxiliary distribution, the attacker's generator would be trapped in this shortcut and fail to explore private information, leading to a synthetic distribution more similar to the public dataset.

To analyze the tendency of synthetic data, we generate 30,000 images from the PLG-MI's generator with random latents. Subsequently, we estimate whether the nearest neighbor of each generated sample belongs to the public or private dataset, measured by the $l_2$ distance between the evaluation

Table 3: Synthetic distribution analysis.

| Attack | Defense | Acc ↑ | AA-1 ↓ | Nearest Neighbor (%) | |
| --- | --- | --- | --- | --- | --- |
| | | | | Public ↑ | Private ↓ |
| GMI* | - | - | - | 73.50 | 26.50 |
| PLG-MI | - | $86.21 \pm 0.91$ | $95.81 \pm 1.63$ | $23.39 \pm 4.27$ | $76.61 \pm 4.27$ |
| | MID | $77.89 \pm 0.70$ | $92.72 \pm 1.64$ | $35.19 \pm 3.51$ | $64.81 \pm 3.51$ |
| | BiDO | $78.97 \pm 0.44$ | $89.18 \pm 1.59$ | $28.99 \pm 3.14$ | $71.01 \pm 3.14$ |
| | NegLS | $81.99 \pm 0.45$ | $89.38 \pm 3.35$ | $22.36 \pm 4.16$ | $77.64 \pm 4.16$ |
| | Trap-MID | $81.37 \pm 1.04$ | $\mathbf{6.23 \pm 5.60}$ | $\mathbf{70.33 \pm 0.40}$ | $\mathbf{29.67 \pm 0.40}$ |

\* The generator is trained independently from target model.

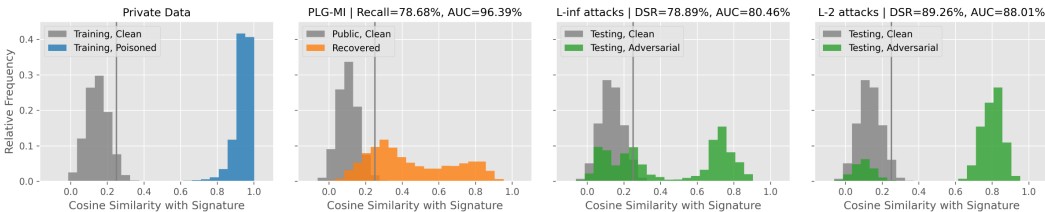

Figure 3: Illustration of trapdoor detection.

model's penultimate outputs. Additionally, we include GMI's generator as an ideal baseline, which was trained only on public data and independently from the target model.

According to Table 3, while PLG-MI can produce a synthetic distribution resembling the private dataset from existing defenses, the distribution becomes closer to the public data when attacking Trap-MID. Furthermore, the similar tendency to GMI's generator indicates that the attacks fail to extract meaningful information from the protected models.

**Trapdoor Recovery Analysis.** To verify the effectiveness of Trap-MID in misleading MI attacks, we assess the presence of trapdoor triggers in the recovered images using a detection method in [20]. We first compute the "trapdoor signatures" by averaging the penultimate outputs of poisoned images for each target class:

$$S_y = \mathbb{E}_{x \sim p(X)}[g_\theta(\Pi_y(X))], \tag{6}$$

where $g_\theta : \mathbb{R}^{d_x} \to \mathbb{R}^{d_z}$ represents the feature extractor of the target model $f_\theta$. After that, we calculate the cosine similarities between benign images and the corresponding trapdoor signature $\cos(g_\theta(X), S_{\hat{y}})$, where $\hat{y}$ is the predicted label of the input. The threshold is then set to be the $k^{th}$ percentile with the desired false positive rate (FPR) $1 - \frac{k}{100}$, and the input data with similarity exceeding the threshold are considered triggered. In this paper, we decide the threshold to achieve a desired FPR of 5%. In addition, since computing each signature with the entire training dataset is computationally expensive, we randomly sample the target label for each training data to estimate the signatures.

As shown in Figure 3, 78.68% of reconstructed images from PLG-MI are reported to be triggered images, indicating that the MI attacks are misled into extracting trapdoor information. Appendix E.6 presents the analysis of other MI attacks.

**Adversarial Detection.** As Shan et al. [20] showed that integrating trapdoors into the model can help detect adversarial attacks, we also assess the effectiveness of Trap-MID in detecting adversarial examples. We utilize AutoAttack [37] and apply the same detection method in the trapdoor recovery analysis. As depicted in Figure 3, Trap-MID achieves a detection success rate (DSR) of over 78% on both $l_\infty$ ($\epsilon = 8/255$) and $l_2$ ($\epsilon = 0.5$) attacks, while maintaining privacy preservation.

Table 4: Adaptive attacks against Trap-MID, using VGG-16 models.

| Aux. Data | Attack | AA-1 $\downarrow$ | AA-5 $\downarrow$ | KNN Dist $\uparrow$ | FID $\uparrow$ |
|---|---|---|---|---|---|
| CelebA | PLG-MI | $6.23 \pm 5.60$ | $13.15 \pm 10.30$ | $2055.96 \pm 147.67$ | $57.82 \pm 23.41$ |
| | PLG-MI++ | $70.44 \pm 35.16$ | $78.14 \pm 36.71$ | $1399.84 \pm 382.13$ | $27.17 \pm 18.03$ |
| FFHQ | PLG-MI | $0.86 \pm 0.39$ | $2.85 \pm 1.27$ | $2227.09 \pm 59.16$ | $94.57 \pm 13.41$ |
| | PLG-MI++ | $31.03 \pm 22.76$ | $44.09 \pm 29.77$ | $1789.95 \pm 324.38$ | $38.98 \pm 31.94$ |

**Adaptive Attacks.** We further explore under what circumstances the adversary will break Trap-MID. Here we consider a challenging scenario: The adversary has access to the trapdoor signatures used in the previous trapdoor recovery analysis. We modify PLG-MI to conduct adaptive attacks, denoted by PLG-MI++. Appendix E.7 demonstrates the scenario where the adversary only knows the existence of trapdoors without information about trapdoor signatures.

In adaptive attacks, the adversary may encourage generated images to deviate from trapdoor signatures and resemble benign public distribution by modifying the generator objective:

$$
\begin{aligned}
\mathcal{L}_{G++} = \mathcal{L}_G - \lambda_{\text{aux}} \mathbb{E}_{Y \sim p_{\text{aux}}(Y)} \big[ \mathbb{E}_{Z \sim p_G(Z)} [\cos(g_\theta(T_{\text{attack}}(G(Z, Y))), S_{\text{aux}, Y})] \big] \\
+ \lambda_{\text{trap}} \mathbb{E}_{Y \sim p_{\text{aux}}(Y)} \big[ \mathbb{E}_{Z \sim p_G(Z)} [\cos(g_\theta(T_{\text{attack}}(G(Z, Y))), S_{\hat{y}})] \big],
\end{aligned}
\tag{7}
$$

where $p_{\text{aux}}(Y)$ denotes the auxiliary distribution of pseudo-labels assigned by PLG-MI's selection strategy, $p_G(Z)$ is the generator's latent distribution, $\mathcal{L}_G$ is the original generator loss, $G : \mathbb{R}^{d_z} \to \mathbb{R}^{d_x}$ is the generator, $T_{\text{attack}} : \mathbb{R}^{d_x} \to \mathbb{R}^{d_x}$ is the random image augmentation used in attacks, $\hat{y}$ is the predicted label of the generated image $G(Z, Y)$, and $\lambda_{\text{aux}}, \lambda_{\text{trap}}$ are the weighting parameters. We set $\lambda_{\text{aux}} = \lambda_{\text{trap}} = 10$, as we found it generally provides a better attack performance. $S_{\text{aux}, y}$ is the auxiliary signature of the target class $y$, computed from public samples:

$$
S_{\text{aux}, y} = \mathbb{E}_{x \sim p_{\text{aux}}(X | y)} [g_\theta(X)],
\tag{8}
$$

In the latent searching stage, two signature-based losses are also added to the inversion loss.

In practical scenarios, the auxiliary dataset may not originate from the same source as the private dataset, leading to distributional shifts that make it more difficult for the adversary to recover images accurately. To demonstrate this case, we also include the experiments with FFHQ [38] as the auxiliary dataset. Appendix E.5 presents more experiments about distributional shifts, where PLG-MI still achieves 89% attack accuracy on the unprotected model.

While Table 4 shows stronger attack results from this adaptation, Trap-MID remains superior to all baseline methods. Additionally, when distributional shifts occur in the auxiliary dataset, the attack performance against Trap-MID degrades significantly. This suggests that the success of MI attacks against Trap-MID, even with adaptive modifications, relies heavily on the similarity between the auxiliary and private data. Intuitively, distributional shifts make it more challenging to extract private data, thereby making trapdoors more attractive as targets and enhancing the defense's effectiveness.

## 5 Conclusion

MI attacks pose significant privacy risks to DNNs' training datasets. Despite existing defense efforts, recent attacks continue to exploit vulnerabilities in these defenses. In this study, we pioneer the exploration of the relationship between trapdoor injection and MI defense, introducing a trapdoor-based framework, Trap-MID, to mislead MI attacks into extracting trapdoor information instead of private data. Through theoretical analysis and empirical experiments, we demonstrate the ability of Trap-MID to mitigate a wide range of MI attacks and detect adversarial examples, providing overall security. Notably, Trap-MID achieves these results without the need for shadow attacks or extra datasets, making it both computationally and data-efficient.

## Acknowledgments and Disclosure of Funding

This work was supported in part by the National Science and Technology Council under Grants NSTC 112-2634-F-002-006, MOST 110-2222-E-002-014-MY3, NSTC 113-2222-E-002-004-MY3, NSTC 113-2923-E-002-010-MY2, NSTC 113-2634-F-002-001-MBK, and by the Center of Data Intelligence: Technologies, Applications, and Systems, National Taiwan University under Grant NTU-113L900903.

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

# A  Broader Impacts, Limitations, and Future Works

## A.1  Broader Impacts

Deep learning has been widely employed in diverse domains and tasks. However, the growing threat of privacy breaches, such as MI attacks, poses significant risks to sensitive data used for model training. Our proposed framework, Trap-MID, offers a promising defense strategy against MI attacks by misleading their exploration directions. Empirical experiments demonstrate its state-of-the-art defense performance. Importantly, Trap-MID achieves these results without the need for additional public datasets or conducting shadow attacks, making it applicable across diverse applications.

## A.2  Limitations and Future Works

### A.2.1  Experimental Limitations

In our empirical experiments, we did not exhaust hyper-parameter tuning via grid search due to the computational constraints. While we found that the configurations in Section 4.1 generally provide a good accuracy-privacy trade-off against various MI attacks, conducting more comprehensive hyper-parameter optimization could further improve utility and defense performance. Ablation studies exploring the impact of different hyper-parameter settings are discussed in Appendix D. Although we did not include the experiments against all MI attacks due to computational requirements, we verified the efficacy of Trap-MID against various white-box attacks [3, 5–8] and a label-only attack [11], in both low-resolution [3, 5, 7, 8, 11] and high-resolution scenarios [6]. These results suggest that Trap-MID is effective across a broad range of MI attacks.

Additionally, we recognize that the attack accuracy metric, based solely on an evaluation classifier trained on the private dataset, may fail to detect out-of-distribution samples, resulting in high KNN distance or FID alongside high attack accuracy. While feature-based metrics like KNN distance and FID can help identify these failures, developing a universal evaluation method could better quantify both attack and defense performance, offering a more straightforward basis for comparison.

### A.2.2  Further Improvements in Trap-MID Design

In this paper, we demonstrate the efficacy of our trapdoor-based framework with a simple trigger design. While we conducted the model training and attacks multiple times to ensure reproducibility, we acknowledge a larger variation in our defense performance than previous defenses due to the randomly initialized triggers. We leave further customization for a more stable and powerful trigger for future work.

In addition, while Trap-MID is more computationally efficient than existing misleading-based defenses, it requires longer training time compared to methods like MID, BiDO, and NegLS due to its three gradient updates per epoch. Developing a more efficient trigger generation process would be a valuable future direction to make Trap-MID more practical for large-scale applications.

### A.2.3  Exploring Different Scenarios

Despite its effectiveness, Trap-MID inherits certain assumptions and limitations from previous works. For instance, it assumes a level of trust between data providers and the model owner to protect their private information [14–16]. A promising future direction involves developing trapdoor injection methods that empower dataset owners or even individual identities to secure sensitive information before sharing data. Additionally, Trap-MID's efficacy may be limited against MI attacks involving KD, such as LOMMA, due to the inherent weaknesses of backdoor attacks. Therefore, future efforts should focus on integrating more robust trapdoors to address these limitations.

In addition, similar to previous works, we used relatively simple victim models for facial recognition, trained on the CelebA dataset with cross-entropy loss. Given the advancements in facial recognition, it would be valuable to evaluate the performance of MI attacks and defenses on more advanced techniques, such as ArcFace [39], or on datasets featuring more diverse poses and facial images in the wild, like IJB-C dataset [40].

Finally, our research represents the first attempt to establish the connection between trapdoor injections and MI defense mechanisms. Extending this defense to different modalities, such as language, graph,

and tabular, or exploring its relationship with other reconstruction-based attacks, such as Gradient Inversion Attacks and Embedding Inversion Attacks, presents an interesting direction for further research.

## B  Proof of Theorem 1

*Proof.* According to Definition 1, for a given model $f$, target label $y$ and corresponding trigger injection function $\Pi_y(\cdot)$, we have:

$$
\begin{aligned}
&T_f(y, \Pi_y) - U_f(y) \\
&= \mathbb{E}_{X \sim p(X)}[\log p_f(y|\Pi_y(X)) - \log p_f(y)] - \mathbb{E}_{X \sim p(X|y)}[\log p_f(y|X) - \log p_f(y)] \\
&= \mathbb{E}_{X \sim p(X)}[\log p_f(y|\Pi_y(X))] - \mathbb{E}_{X \sim p(X|y)}[\log p_f(y|X)] \\
&\geq \delta.
\end{aligned}
\tag{9}
$$

Expanding the KL divergence $D_{KL}(p(X)||p(\Pi(X)))$ in Definition 2:

$$
D_{KL}(p(X)||p(\Pi(X))) = \mathbb{E}_{X \sim p(X)}[\log p(X) - \log p(\Pi(X))] \leq \epsilon.
\tag{10}
$$

As a result, if for all $y$, the trapdoor is $(\delta, y)$-effective and $\epsilon$-natural on the model $f$ with injection function $\Pi_y(\cdot)$, we have:

$$
\begin{aligned}
&\mathbb{E}_{Y \sim p(Y)}\mathbb{E}_{X \sim p(X)}[\log p_f(\Pi_y(X)|Y)] \\
&= \mathbb{E}_{Y \sim p(Y)}\mathbb{E}_{X \sim p(X)}[\log p_f(Y|\Pi_y(X)) + \log p(\Pi_y(X)) - \log p_f(Y)] \\
&\geq \mathbb{E}_{Y \sim p(Y)}\Big[\mathbb{E}_{X \sim p(X|Y)}[\log p_f(Y|X)] + \delta\Big] + \Big(\mathbb{E}_{X \sim p(X)}[\log p(X)] - \epsilon\Big) - \mathbb{E}_{Y \sim p(Y)}[\log p_f(Y)] \\
&= \mathbb{E}_{(X,Y) \sim p(X,Y)}[\log p_f(Y|X) + \log p(X) - \log p_f(Y)] + (\delta - \epsilon) \\
&= \mathbb{E}_{(X,Y) \sim p(X,Y)}[\log p_f(X|Y)] + (\delta - \epsilon).
\end{aligned}
\tag{11}
$$

$\square$

## C  Experimental Details

### C.1  Hardware and Software Details

All experiments were conducted on an Intel Xeon Gold 6226R CPU with an NVIDIA RTX A6000 GPU. The average execution time of Trap-MID training is 1 hour 15 minutes with a standard deviation of 13 seconds. Our source code is publicly available at `https://github.com/ntuaislab/Trap-MID` to reproduce main experiments.

### C.2  Datasets

The datasets we used in our experiments are all publicly accessible, including:

**CelebA.**  CelebA [30] comprises 202,599 facial images of 10,177 identities with coarse alignment. Following prior studies, we selected the 1,000 identities with the most samples as the private dataset, totaling 30,029 facial images. The private dataset was then divided into training and testing datasets for utility evaluation, containing 27,018 and 3,009 samples, respectively. For the auxiliary dataset used in MI attacks, a disjoint subset of the CelebA dataset was sampled, which contains 30,000 images without overlapping identities with the private dataset. Images are cropped at the center and resized to $64 \times 64$ pixels.

**FFHQ.**  FFHQ [38] contains 70,000 high-quality facial images with considerable variation in age, ethnicity, and background. The entire FFHQ dataset was utilized as the adversary's auxiliary dataset in the MI attacks with distributional shifts.

---
**Algorithm 1** Training procedure for Trap-MID
---
**Input:** Training dataset $\{(x_i, y_i)\}_{i=1}^{N}$, a classifier $f_\theta$ parameterized by $\theta$, a discriminator $D_\phi$ parameterized $\phi$, trapdoor triggers $K = \{k_i\}_{i=1}^{d_y}$ corresponding with different classes, mini-batch size $m$, number of mini-batches $M$, learning rate $\alpha$, trigger step size $\epsilon$, and training epoch $T$
**Output:** A trained model with privacy protection
   **for** $t \leftarrow 1, \ldots, T$ **do**
      **for** $j \leftarrow 1, \ldots, M$ **do**
         Sample a mini-batch $\{(x_i, y_i)\}_{i=1}^{m}$ from training dataset.
         Sample a set of target labels $\{y_i'\}_{i=1}^{m}$ with the same size of the mini-batch.
         $\phi \leftarrow \phi - \alpha \nabla \mathcal{L}_D$.                                 $\triangleright$ Equation 4
         $K \leftarrow K - \epsilon \operatorname{sgn}(\nabla \mathcal{L}_{\text{trigger}})$.                $\triangleright$ Equation 5
         Clip the triggers $K$ into the range $[0, 1]$.
         $\theta \leftarrow \theta - \alpha \nabla \mathcal{L}_\theta$.                               $\triangleright$ Equation 2
      **end for**
   **end for**
---

## C.3 Target Models

Consistent with prior research, we trained the VGG-16 and Face.evoLVe models for 50 epochs, and the ResNet-152 models for 40 epochs. All models were trained using the SGD optimizer with a batch size of 64, a learning rate of 0.01, a momentum value of 0.9, and a weight decay value of 0.0001.

## C.4 Trap-MID

For Trap-MID, we adopted the same hyper-parameters as the unprotected model. Algorithm 1 outlines the training process. The protected model was configured to achieve a trapdoor success rate exceeding 99%. Typically, we used a blend ratio $\alpha = 0.02$ and a trapdoor loss weight $\beta = 0.2$. Trapdoor triggers were randomly initialized using a uniform distribution within $[0, 1]$ and then updated with a step size $\epsilon = 0.01$. The discriminator was optimized with identical settings to the target classifier.

The random augmentation pipeline consisted of a sequence of image transformations, including random resized crops with cropping scales sampled from $[0.8, 1]$, horizontal flips, and random rotations with degrees sampled from $[-30, 30]$. Each augmentation was applied randomly with a probability of 0.5. Note that our augmentation strategy differs from that in MI attacks to prevent the requirements of attack information. However, since these augmentations are widely adopted across various domains, our approach still overlaps with those utilized in MI attacks.

## C.5 Attack Methods

We conducted MI attacks using the official implementation of GMI, KED-MI, LOMMA, and PLG-MI. Specifically, we utilized the PLG-MI official code available from `https://github.com/LetheSec/PLG-MI-Attack` for GMI, KED-MI, and PLG-MI attacks. For LOMMA, we employed the official code available at `https://github.com/sutd-visual-computing-group/Re-thinking_MI`. During the latent searching stage, we optimized the latent for 1,500 epochs in GMI, KED-MI, and LOMMA, and 600 epochs in PLG-MI.

## C.6 Baseline Defenses

**MID.** The protected model was trained with the MID official code provided at `https://github.com/Jiachen-T-Wang/mi-defense`. For the Face.evoLVe and ResNet-152 models, we added the information bottleneck before the final fully connected layer, with a bottleneck size of 512. The mutual information was approximated using the same variational method as the official implementation. The weight coefficient of the regularization term $\lambda$ was set to 0.003.

**BiDO.** We utilized the BiDO official code at `https://github.com/AlanPeng0897/Defend_MI` to train the protected models. The models were trained with Adam optimizer, using a learning rate of 0.0001 without weight decay. For Face.evoLVe and ResNet-152 models, we used the latent representations from the four major ResNet blocks to estimate the bilateral dependency. Typically,

the Hilbert-Schmidt Independence Criterion (HSIC) was adopted as the dependency measure, as it was reported with a better defense performance than the Constrained Covariance (COCO) in [15]. The balancing hyper-parameters $(\lambda_x, \lambda_y)$ were set to $(0.05, 0.5)$.

**NegLS.** Since the NegLS code was not available when we conducted our experiments, we adapted our implementation to include their negative label smoothing based on the official configuration. The models were trained for 100 epochs using the Adam optimizer with a batch size of 128 and an initial learning rate of 0.001 without weight decay. The learning rate was then multiplied by a factor of 0.1 at the $75^{th}$ and $90^{th}$ epochs. The label smoothing factor $\alpha$ was set to be 0 at the first 50 epochs. During the $51^{th}$ to the $75^{th}$ epoch, $\alpha$ was set to be $-0.05 \times \frac{t-50}{100-50}$, where $t$ is the current epoch. After that, $\alpha$ was fixed to be $-0.05$ for the remaining training.

## C.7 Evaluation Models

An evaluation model with a different architecture from the target model is trained on the same private dataset, acting as an additional observer to assess the success of MI attacks. We use the evaluation model from [8], which is a Face.evoLVe model [32] pre-trained on MS-Celeb1M [41] and fine-tuned on the private training dataset. The evaluation model has an input resolution of 112x112, with the 64x64 images resized to fit its input size. It achieves a 95.88% accuracy on the testing dataset.

## C.8 Evaluation Metrics

**Attack accuracy (AA).** For each recovered image, we determine whether it is classified as the target class by the evaluation model, yielding both top-1 and top-5 attack accuracy.

**K-Nearest Neighbor Distance (KNN Dist).** The KNN distance calculates the shortest $l_2$ distance from the reconstructed images to the private data in the feature space of the evaluation model. Specifically, to demonstrate the success of reconstruction, each recovered image is compared only with the private data belonging to its target class.

**Fréchet Inception Distance (FID).** The FID is evaluated by the difference in means and covariances between the generated and real images in the feature space of Inception-v3 [42]. Consistent with prior studies, we compute FID only on the successfully recovered images identified by the evaluation classifier to measure the quality and diversity of the extracted information.

# D Ablation Studies

In this section, we first assess the effectiveness of TeD's trapdoor injection strategy [20] against MI attacks, and then analyze the impact of various configurations on defense performance.

Since conducting the whole experiment multiple times would be computationally expensive, without specification, we use the same evaluation protocol in previous works for the experiments in the appendices. Typically, we train a single target model for each configuration and conduct MI attacks to reconstruct 5 images per class with random initialization in the latent searching stage. The standard deviation of attack accuracy is then computed from 5 reconstruction attempts. For experiments sharing the same setups as those in Section 4.1, we average the evaluation metrics and their standard deviations across multiple runs.

## D.1 TeD's Trapdoor Injection Strategy

To verify whether TeD's protected models inherently defend against MI attacks, we employ the same trapdoor injection method in [20] and leave other training setups consistent with the unprotected model. Specifically, each trapdoor trigger comprises five $(6 \times 6)$-pixel squares randomly scattered across the image, with a blend ratio $\alpha = 0.1$ or $0.2$ and a trapdoor loss weight $\beta = 0.5$.[2] The intensity of each square was sampled from $\mathcal{N}(\mu, \sigma)$ with uniformly sampled $\mu \in [0, 1]$ and $\sigma \in [0, 1]$, and was fixed during model training. Sample poisoned images are shown in Figure 4.

---

[2]The blend ratio $\alpha$ and the trapdoor loss weight $\beta$ in our paper are equivalent to the mask ratio $\kappa$ and the injection ratio $\lambda$ in [20].

**TeD**  **Trap-MID** 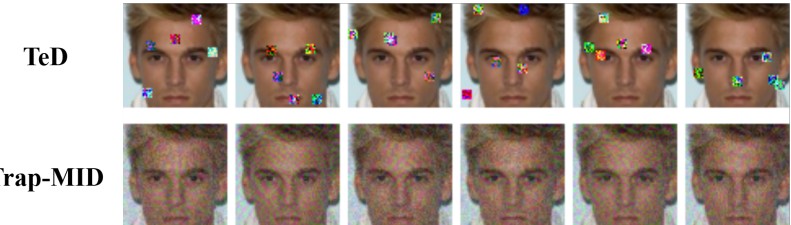

Figure 4: Sample poisoned images for TeD's ($\alpha = 0.1$) and our trapdoor injection methods. Each column depicts a poisoned image with a specific target label. The blend ratio $\alpha$ is multiplied by a factor of 10 for better visualization ($\alpha = 1$ and 0.2 for TeD's and our triggers, respectively).

For further comparison, we adopt the same configurations of Trap-MID, such as a smaller blend ratio ($\alpha = 0.02$), a smaller trapdoor loss weight ($\beta = 0.2$), trigger optimization and data augmentation, while keeping TeD's five-square patterns. This enhanced version is denoted as TeD+.

Table 5 compares defense performances using TeD's trapdoor injection techniques. Although decreasing the blend ratio and adopting our configurations can improve defense performance, TeD models remain vulnerable to PLG-MI with their five-square patterns. In contrast, spreading the trapdoor information across all image pixels is crucial for fooling stronger MI attacks. Intuitively, while TeD's patch-based triggers fit the $l_\infty$ or $l_2$ budgets in adversarial attacks [20], the resulting poisoned images appear less natural, making them more likely to be identified and penalized by the attacker's discriminator. Moreover, as TeD's triggers highly rely on local information around specific locations, they may exhibit lower robustness against spatial transformations, leading to inferior defense against PLG-MI.

Table 5: Defense comparison with TeD's trapdoors, using VGG-16 models.

| Defense | Acc ↑ | AA-1 ↓ | AA-5 ↓ | KNN Dist ↑ | FID ↑ |
|---|---|---|---|---|---|
| GMI | | | | | |
| - | 86.21 | 14.29 ± 2.43 | 32.64 ± 2.77 | 1798.23 | 31.01 |
| TeD ($\alpha = 0.2$) | 85.01 | 10.84 ± 2.19 | 26.22 ± 3.09 | 1869.42 | 34.09 |
| TeD ($\alpha = 0.1$) | 83.98 | 1.28 ± 0.61 | 4.64 ± 1.12 | 2285.70 | 73.25 |
| TeD+ | 81.85 | 1.58 ± 0.83 | 5.28 ± 1.10 | 2217.72 | 57.55 |
| Trap-MID* | 81.37 | **0.24 ± 0.27** | **1.16 ± 0.62** | **2411.39** | **153.73** |
| KED-MI | | | | | |
| - | 86.21 | 56.46 ± 2.23 | 82.84 ± 1.96 | 1404.85 | 17.10 |
| TeD ($\alpha = 0.2$) | 85.01 | 47.78 ± 2.02 | 73.98 ± 1.78 | 1453.73 | 16.27 |
| TeD ($\alpha = 0.1$) | 83.98 | 28.64 ± 2.00 | 48.80 ± 1.95 | 1706.64 | 18.38 |
| TeD+ | 81.85 | 19.78 ± 1.30 | 39.78 ± 2.32 | 1771.91 | 23.72 |
| Trap-MID* | 81.37 | **9.24 ± 1.15** | **19.24 ± 1.31** | **2056.00** | **87.39** |
| LOMMA (GMI) | | | | | |
| - | 86.21 | 67.60 ± 5.71 | 88.96 ± 3.91 | 1414.00 | 38.94 |
| TeD ($\alpha = 0.2$) | 85.01 | 65.22 ± 5.76 | 89.50 ± 3.84 | 1428.35 | **39.33** |
| TeD ($\alpha = 0.1$) | 83.98 | 53.34 ± 5.32 | 80.60 ± 4.08 | 1529.23 | 40.38 |
| TeD+ | 81.85 | 48.96 ± 6.23 | 75.40 ± 4.79 | 1532.89 | 38.57 |
| Trap-MID* | 81.37 | **41.63 ± 5.61** | **68.24 ± 5.33** | **1569.92** | 39.29 |
| LOMMA (KED-MI) | | | | | |
| - | 86.21 | 79.47 ± 3.93 | 95.16 ± 2.06 | 1279.48 | 22.70 |
| TeD ($\alpha = 0.2$) | 85.01 | 75.62 ± 4.41 | 93.50 ± 2.31 | 1295.36 | 20.38 |
| TeD ($\alpha = 0.1$) | 83.98 | 73.18 ± 4.11 | 93.80 ± 2.52 | 1328.78 | 21.86 |
| TeD+ | 81.85 | 63.72 ± 4.48 | 87.50 ± 3.29 | 1395.46 | **26.04** |
| Trap-MID* | 81.37 | **61.25 ± 4.28** | **85.76 ± 3.26** | **1404.77** | 24.19 |
| PLG-MI | | | | | |
| - | 86.21 | 95.81 ± 1.54 | 99.43 ± 0.58 | 1174.13 | 12.77 |
| TeD ($\alpha = 0.2$) | 85.01 | 95.86 ± 1.90 | 99.30 ± 0.56 | 1163.27 | 12.44 |
| TeD ($\alpha = 0.1$) | 83.98 | 93.10 ± 2.15 | 98.60 ± 1.13 | 1219.55 | 15.96 |
| TeD+ | 81.85 | 90.94 ± 2.26 | 97.90 ± 1.24 | 1174.40 | 15.80 |
| Trap-MID* | 81.37 | **6.23 ± 1.70** | **13.15 ± 2.57** | **2055.96** | **57.82** |

* The mean and standard deviation of each evaluation metric are averaged across multiple runs.

Table 6: Defense comparison with trigger optimization, using VGG-16 models.

| Defense | Acc ↑ | AA-1 ↓ | AA-5 ↓ | KNN Dist ↑ | FID ↑ |
|---|---|---|---|---|---|
| GMI | | | | | |
| Trap-MID (fixed triggers) | 76.03 | 0.12 ± 0.23 | **0.36 ± 0.42** | **2507.48** | 141.77 |
| + Trapdoor loss | 83.38 | **0.04 ± 0.11** | 0.44 ± 0.40 | 2481.64 | **186.14** |
| + Discriminator* | 81.37 | 0.24 ± 0.27 | 1.16 ± 0.62 | 2411.39 | 153.73 |
| KED-MI | | | | | |
| Trap-MID (fixed triggers) | 76.03 | **0.08 ± 0.22** | **0.74 ± 0.51** | **2397.43** | **269.08** |
| + Trapdoor loss | 83.38 | 0.50 ± 0.36 | 1.72 ± 0.75 | 2339.31 | 96.66 |
| + Discriminator* | 81.37 | 9.24 ± 1.15 | 19.24 ± 1.31 | 2056.00 | 87.39 |
| LOMMA (GMI) | | | | | |
| Trap-MID (fixed triggers) | 76.03 | **39.94 ± 4.95** | **66.40 ± 5.90** | **1592.95** | **41.86** |
| + Trapdoor loss | 83.38 | 41.54 ± 5.57 | 67.00 ± 4.17 | 1581.92 | 41.82 |
| + Discriminator* | 81.37 | 41.63 ± 5.61 | 68.24 ± 5.33 | 1569.92 | 39.29 |
| LOMMA (KED-MI) | | | | | |
| Trap-MID (fixed triggers) | 76.03 | **48.72 ± 4.61** | **77.30 ± 3.57** | **1471.80** | **27.59** |
| + Trapdoor loss | 83.38 | 63.62 ± 4.36 | 87.60 ± 3.40 | 1393.89 | 24.19 |
| + Discriminator* | 81.37 | 61.25 ± 4.28 | 85.76 ± 3.26 | 1404.77 | 24.19 |
| PLG-MI | | | | | |
| Trap-MID (fixed triggers) | 76.03 | 90.92 ± 2.36 | 97.78 ± 1.24 | 1182.63 | 17.03 |
| + Trapdoor loss | 83.38 | 89.44 ± 2.20 | 97.56 ± 1.06 | 1258.22 | 18.57 |
| + Discriminator* | 81.37 | **6.23 ± 1.70** | **13.15 ± 2.57** | **2055.96** | **57.82** |

* The mean and standard deviation of each evaluation metric are averaged across multiple runs.

## D.2 Trigger Optimization

Table 6 demonstrates the impact of trigger optimization. Fixed triggers generally preserve privacy against most MI attacks, except for PLG-MI, but result in a noticeable accuracy reduction. In contrast, the influence of trapdoor loss and discriminator loss varies based on the capacity of the GAN used in the attacks.

**Trapdoor loss.** Incorporating trapdoor loss can create easily learnable triggers for the target model, reducing accuracy drop. However, these crafted adversarial-like triggers may be more difficult to generate, resulting in lower defense performance against attacks using weaker generators like GMI, KED-MI, and LOMMA.

**Discriminator loss.** Discriminator loss promotes invisible triggers, which are crucial for deceiving attacks with stronger discriminators, such as PLG-MI. However, generating invisible triggers requires fine-grain adjustments, making them less effective against attacks using weaker generators.

Table 7: Defense comparison with different blend ratios, using VGG-16 models.

| Defense | Acc ↑ | AA-1 ↓ | AA-5 ↓ | KNN Dist ↑ | FID ↑ |
|---|---|---|---|---|---|
| GMI | | | | | |
| Trap-MID ($\alpha = 0.1$) | 83.18 | $1.36 \pm 0.81$ | $4.58 \pm 1.44$ | 2223.08 | 62.26 |
| Trap-MID ($\alpha = 0.05$) | 84.14 | $0.52 \pm 0.38$ | $1.72 \pm 0.65$ | 2413.40 | 113.84 |
| Trap-MID ($\alpha = 0.03$) | 83.08 | $\mathbf{0.10 \pm 0.22}$ | $\mathbf{0.58 \pm 0.73}$ | $\mathbf{2470.07}$ | 136.72 |
| Trap-MID ($\alpha = 0.02$)* | 81.37 | $0.24 \pm 0.27$ | $1.16 \pm 0.62$ | 2411.39 | $\mathbf{153.73}$ |
| KED-MI | | | | | |
| Trap-MID ($\alpha = 0.1$) | 83.18 | $34.54 \pm 1.85$ | $63.22 \pm 1.64$ | 1537.62 | 19.29 |
| Trap-MID ($\alpha = 0.05$) | 84.14 | $35.06 \pm 1.95$ | $60.38 \pm 2.28$ | 1594.44 | 23.23 |
| Trap-MID ($\alpha = 0.03$) | 83.08 | $\mathbf{6.04 \pm 1.29}$ | $\mathbf{12.66 \pm 1.28}$ | $\mathbf{2111.42}$ | 66.57 |
| Trap-MID ($\alpha = 0.02$)* | 81.37 | $9.24 \pm 1.15$ | $19.24 \pm 1.31$ | 2056.00 | $\mathbf{87.39}$ |
| LOMMA (GMI) | | | | | |
| Trap-MID ($\alpha = 0.1$) | 83.18 | $50.54 \pm 5.77$ | $76.00 \pm 4.63$ | 1513.98 | 33.46 |
| Trap-MID ($\alpha = 0.05$) | 84.14 | $39.30 \pm 5.76$ | $66.40 \pm 5.93$ | 1588.70 | 38.88 |
| Trap-MID ($\alpha = 0.03$) | 83.08 | $\mathbf{36.62 \pm 5.51}$ | $\mathbf{65.10 \pm 4.81}$ | $\mathbf{1614.48}$ | $\mathbf{40.71}$ |
| Trap-MID ($\alpha = 0.02$)* | 81.37 | $41.63 \pm 5.61$ | $68.24 \pm 5.33$ | 1569.92 | 39.29 |
| LOMMA (KED-MI) | | | | | |
| Trap-MID ($\alpha = 0.1$) | 83.18 | $70.22 \pm 4.53$ | $89.70 \pm 2.48$ | 1344.36 | 23.01 |
| Trap-MID ($\alpha = 0.05$) | 84.14 | $71.14 \pm 3.81$ | $91.20 \pm 3.01$ | 1327.49 | 24.53 |
| Trap-MID ($\alpha = 0.03$) | 83.08 | $65.46 \pm 4.28$ | $88.50 \pm 2.70$ | 1404.52 | $\mathbf{25.43}$ |
| Trap-MID ($\alpha = 0.02$)* | 81.37 | $\mathbf{61.25 \pm 4.28}$ | $\mathbf{85.76 \pm 3.26}$ | $\mathbf{1404.77}$ | 24.19 |
| PLG-MI | | | | | |
| Trap-MID ($\alpha = 0.1$) | 83.18 | $89.40 \pm 2.03$ | $97.20 \pm 1.35$ | 1219.49 | 18.49 |
| Trap-MID ($\alpha = 0.05$) | 84.14 | $93.86 \pm 1.93$ | $98.94 \pm 0.83$ | 1187.61 | 13.67 |
| Trap-MID ($\alpha = 0.03$) | 83.08 | $\mathbf{1.92 \pm 1.12}$ | $\mathbf{5.48 \pm 1.79}$ | $\mathbf{2185.82}$ | $\mathbf{60.35}$ |
| Trap-MID ($\alpha = 0.02$)* | 81.37 | $6.23 \pm 1.70$ | $13.15 \pm 2.57$ | 2055.96 | 57.82 |

\* The mean and standard deviation of each evaluation metric are averaged across multiple runs.

## D.3 Blend Ratio

In this section, we tune the blend ratio $\alpha$ within $[0.02, 0.1]$ and compare the defense performance. As shown in Table 7, decreasing the blend ratio generally makes the trapdoor triggers more invisible and improves defense performance. Notably, there is a sharp drop in attack accuracy. For instance, PLG-MI's attack accuracy drops from 93.86% to 1.92% when the blend ratio is reduced from 0.05 to 0.03, suggesting that an adequately invisible trigger is crucial for misleading certain attacks. In terms of model utility, Trap-MID is relatively insensitive to the blend ratio, maintaining about 81-84% testing accuracy across different configurations, demonstrating its effectiveness without a significant accuracy loss.

Table 8: Defense comparison with different trapdoor loss weight, using VGG-16 models.

| Defense | Acc ↑ | AA-1 ↓ | AA-5 ↓ | KNN Dist ↑ | FID ↑ |
|---|---|---|---|---|---|
| GMI | | | | | |
| - | 86.21 | 14.29 ± 2.43 | 32.64 ± 2.77 | 1798.23 | 31.01 |
| Trap-MID ($\beta = 0.02$) | 84.01 | **0.10 ± 0.19** | **0.62 ± 0.55** | **2448.06** | **203.41** |
| Trap-MID ($\beta = 0.05$) | 81.15 | 0.30 ± 0.30 | 1.20 ± 0.63 | 2389.23 | 120.15 |
| Trap-MID ($\beta = 0.1$) | 82.55 | 0.58 ± 0.55 | 2.36 ± 0.93 | 2313.73 | 104.30 |
| Trap-MID ($\beta = 0.2$)* | 81.37 | 0.24 ± 0.27 | 1.16 ± 0.62 | 2411.39 | 153.73 |
| KED-MI | | | | | |
| - | 86.21 | 56.46 ± 2.23 | 82.84 ± 1.96 | 1404.85 | 17.10 |
| Trap-MID ($\beta = 0.02$) | 84.01 | 10.78 ± 0.91 | 23.84 ± 1.90 | 1925.00 | 34.42 |
| Trap-MID ($\beta = 0.05$) | 81.15 | **5.84 ± 1.07** | **14.16 ± 1.48** | 1979.30 | 62.55 |
| Trap-MID ($\beta = 0.1$) | 82.55 | 20.54 ± 2.20 | 41.22 ± 1.57 | 1681.44 | 23.04 |
| Trap-MID ($\beta = 0.2$)* | 81.37 | 9.24 ± 1.15 | 19.24 ± 1.31 | **2056.00** | **87.39** |
| LOMMA (GMI) | | | | | |
| - | 86.21 | 67.60 ± 5.71 | 88.96 ± 3.91 | 1414.00 | 38.94 |
| Trap-MID ($\beta = 0.02$) | 84.01 | 44.46 ± 6.25 | 74.50 ± 5.74 | 1549.12 | **39.81** |
| Trap-MID ($\beta = 0.05$) | 81.15 | 42.42 ± 5.67 | 70.10 ± 5.79 | 1569.07 | 36.18 |
| Trap-MID ($\beta = 0.1$) | 82.55 | 48.84 ± 5.99 | 75.00 ± 4.73 | 1516.63 | 35.87 |
| Trap-MID ($\beta = 0.2$)* | 81.37 | **41.63 ± 5.61** | **68.24 ± 5.33** | **1569.92** | 39.29 |
| LOMMA (KED-MI) | | | | | |
| - | 86.21 | 79.47 ± 3.93 | 95.16 ± 2.06 | 1279.48 | 22.70 |
| Trap-MID ($\beta = 0.02$) | 84.01 | 71.52 ± 3.76 | 91.70 ± 2.54 | 1347.84 | 27.79 |
| Trap-MID ($\beta = 0.05$) | 81.15 | **58.76 ± 4.43** | **85.30 ± 3.46** | **1424.57** | **28.93** |
| Trap-MID ($\beta = 0.1$) | 82.55 | 69.16 ± 4.07 | 90.20 ± 2.76 | 1359.77 | 22.28 |
| Trap-MID ($\beta = 0.2$)* | 81.37 | 61.25 ± 4.28 | 85.76 ± 3.26 | 1404.77 | 24.19 |
| PLG-MI | | | | | |
| - | 86.21 | 95.81 ± 1.54 | 99.43 ± 0.58 | 1174.13 | 12.77 |
| Trap-MID ($\beta = 0.02$) | 84.01 | 23.84 ± 3.54 | 38.72 ± 3.58 | 1736.19 | 29.96 |
| Trap-MID ($\beta = 0.05$) | 81.15 | 30.30 ± 3.28 | 50.52 ± 4.00 | 1665.16 | 18.66 |
| Trap-MID ($\beta = 0.1$) | 82.55 | 10.46 ± 2.12 | 21.74 ± 2.79 | 1937.55 | 28.84 |
| Trap-MID ($\beta = 0.2$)* | 81.37 | **6.23 ± 1.70** | **13.15 ± 2.57** | **2055.96** | **57.82** |

* The mean and standard deviation of each evaluation metric are averaged across multiple runs.

## D.4 Loss Weight

Table 8 illustrates the impact of the trapdoor loss weight. In general, A larger weight prioritizes learning trapdoor behavior over the main task, enhancing defense at the cost of accuracy and leading to an accuracy-privacy trade-off. Notably, Trap-MID achieves privacy protection without a significant accuracy drop, with testing accuracy decreasing from 86.21% to 81.37% when the trapdoor loss weight is increased to 0.2. Moreover, a relatively low weight (e.g., $\beta = 0.02$) is sufficient to reduce the performance of most MI attacks. For example, the top-1 attack accuracy drops from 56.46% to 10.78% against KED-MI, and from 95.81% to 23.84% against PLG-MI.

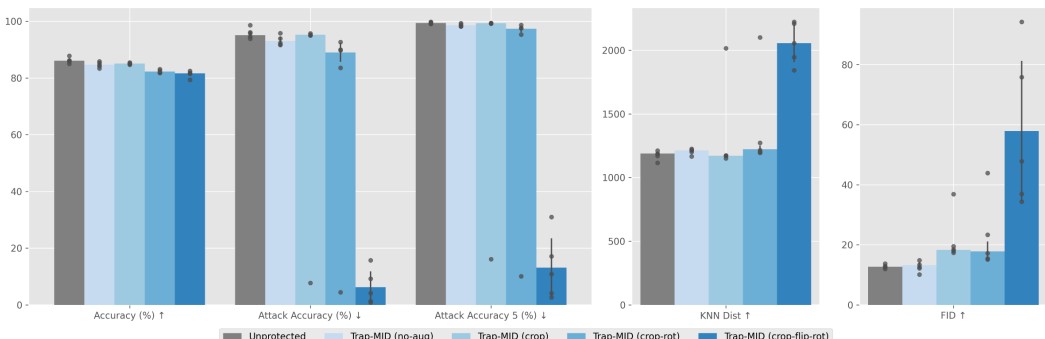

Figure 5: Defense comparison with different augmentation against PLG-MI, using VGG-16 models.

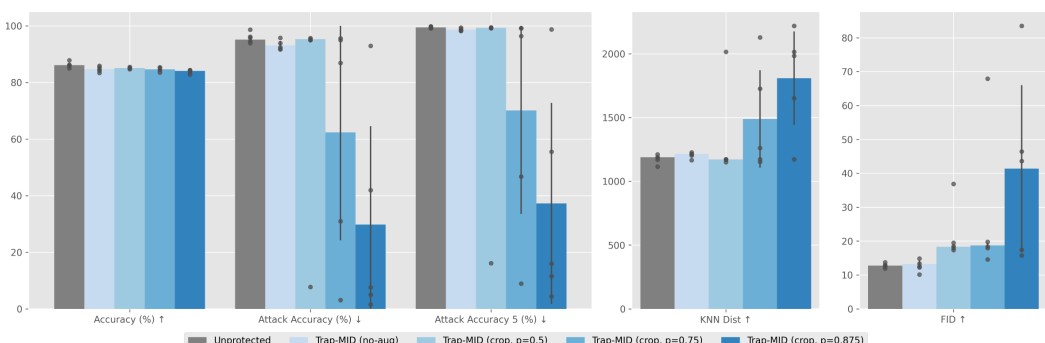

Figure 6: Defense comparison with different augmentation probabilities against PLG-MI, using VGG-16 models.

## D.5 Augmentation

This section investigates the impact of augmentations used in model training. Since we noticed that weak augmentation can lead to unstable defense performance, we employed the same 5-run evaluation protocols as in Section 4.1 in this section. Moreover, the mean and standard deviation of each metric were estimated after removing outliers by the IQR method due to large variation.

Figure 5 shows the defense performance against PLG-MI with different numbers of augmentations. Typically, we started from defense without augmentation and incrementally added random resized crop, random rotation, and horizontal flip. Although protected models with fewer augmentations can sometimes reduce attack performance significantly, they remain vulnerable to PLG-MI in most runs. Intuitively, augmentation helps identify and address the trapdoor's weaknesses in terms of transformation robustness. While it is possible to randomly sample and optimize a robust trigger with weak augmentations, stronger augmentations are more effective at detecting these weaknesses comprehensively, resulting in more stable and improved defense performance.

In addition, to verify whether applying augmentation more frequently can enhance the defense, we increased the probability of applying transformation from 50% to 87.5% using only random resized crop. As Figure 6 demonstrates, even with a single augmentation, applying it frequently can reveal more trapdoors' weaknesses and improve the defense performance of protected models.

Table 9: Training time comparison, using VGG-16 models.

| Defense | Training Time ↓ |
|---|---|
| - | 15 mins |
| MID | 15 mins |
| BiDO | 16 mins |
| NegLS | 35 mins |
| Trap-MID | 1 hour 15 mins |

Table 10: The worst-case performance of Trap-MID compared to the best-case performance of existing defenses, using VGG-16 models.

| Attack | Defense | Acc ↑ | AA-1 ↓ | AA-5 ↓ | KNN Dist ↑ | FID ↑ |
|---|---|---|---|---|---|---|
| GMI | BiDO (best) | 78.32 | 4.42 | 12.94 | 2036.78 | 47.55 |
| | Trap-MID (worst) | 79.39 | **0.56** | **2.46** | **2280.19** | **75.16** |
| KED-MI | NegLS (best) | 81.79 | 29.64 | 57.28 | 1544.90 | **47.31** |
| | Trap-MID (worst) | 81.55 | **23.80** | **46.58** | **1665.86** | 21.23 |
| LOMMA (GMI) | NegLS (best) | 81.79 | 48.58 | 75.80 | 1423.78 | **38.27** |
| | Trap-MID (worst) | 81.55 | **44.80** | **72.60** | **1535.01** | 35.47 |
| LOMMA (KED-MI) | MID (best) | 76.67 | **59.18** | **86.30** | **1413.53** | **24.55** |
| | Trap-MID (worst) | 81.55 | 69.32 | 90.90 | 1333.36 | 20.95 |
| PLG-MI | NegLS (best) | 81.79 | 83.84 | 96.58 | 1495.23 | **73.45** |
| | Trap-MID (worst) | 79.39 | **15.72** | **30.98** | **1843.42** | 36.91 |

# E  Additional Experimental Results

## E.1  Training Time Comparison

Table 9 presents the training time for various defense methods. While Trap-MID requires the longest time due to its three gradient updates per epoch for the discriminator, triggers, and target model, it is worth noting that it significantly outperforms other defenses against recent MI attacks. Furthermore, Trap-MID still requires less data and computational resources compared to existing misleading-based defenses, eliminating the need for an additional dataset, training an extra classifier, or executing shadow attacks.

We believe that enhancing the efficiency of trigger generation would be a valuable future direction, making Trap-MID more practical for large-scale applications. For example, pre-computing triggers with fewer steps may reduce overhead during model training.

## E.2  Worst-case Performance of Trap-MID

In prior experiments, the randomly initialized triggers in Trap-MID introduced variability in defense performance, resulting in a larger standard deviation. This section highlights the worst-case performance of Trap-MID compared to the best-case performance of existing defenses in terms of top-1 attack accuracy. As shown in Table 10, the worst-case performance of Trap-MID surpasses the best-case performance of existing methods against most attacks, demonstrating its effectiveness.

## E.3  Additional Evaluation Metrics

This section further compares the defense performance with additional evaluation metrics used by [4] and [6], including:

Table 11: Defense comparison against PLG-MI, using VGG-16 models and measuring with additional evaluation metrics.

| Defense | Acc ↑ | $\delta_{\text{face}}$ ↑ | Precision ↓ | Recall ↓ | Density ↓ | Coverage ↓ |
|---|---|---|---|---|---|---|
| - | 87.83 | 0.6110 | 19.39 | 13.17 | 0.0893 | 0.1498 |
| MID | 76.67 | 0.6410 | 21.25 | 33.96 | 0.0913 | 0.1734 |
| BiDO | 79.62 | 0.7058 | 20.17 | 10.17 | 0.0807 | 0.1362 |
| NegLS | 81.76 | 0.7587 | **3.80** | **0.00** | **0.0244** | **0.0189** |
| Trap-MID | 81.62 | **1.3845** | 9.56 | 71.63 | 0.0328 | 0.0728 |

**KNN Distance in the FaceNet [43] Feature Space ($\delta_{\text{face}}$).** This metric estimates the FaceNet feature distance between each recovered image and the nearest private data. A higher value indicates less similarity to the private data.

**Improved Precision [44].** This metric assesses whether each recovered image lies within the manifold of private data in the InceptionV3 feature space. A lower value signifies less similarity to the private data.

**Improved Recall [44].** This evaluates whether each private image is encompassed within the manifold of recovered data in the InceptionV3 feature space. A lower value suggests that the generator is less likely to reproduce private data.

**Density [45].** This metric quantifies how many private-sample neighborhood spheres contain each recovered image in the InceptionV3 feature space. A lower value indicates less similarity to the private data.

**Coverage [45].** This assesses how many private samples have a neighborhood sphere that contains at least one recovered image in the InceptionV3 feature space. A lower value suggests that the generator is less likely to reproduce private data.

As shown in Table 11, Trap-MID outperforms existing defenses in FaceNet distance and ranks second to NegLS in most other metrics. For the improved recall metrics, since arbitrary images with injected triggers can be classified into corresponding classes, the recovered samples become more diverse, leading to a broader manifold and a higher recall value. Additionally, all metrics, except for FaceNet distance, utilize the same InceptionV3 model as FID. This gives NegLS an advantage in these metrics due to its less natural recovered images.

### E.4 Defense Performance on Different Architectures

In this section, we present the defense comparison on the Face.evoLVe and ResNet-152 models. As shown in Table 12 and Table 13, Trap-MID consistently outperforms existing defenses on both Face.evoLVe and ResNet-152 models.

Table 12: Defense comparison on Face.evoLVe models.

| Attack | Defense | Acc ↑ | AA-1 ↓ | AA-5 ↓ | KNN Dist ↑ | FID ↑ |
|---|---|---|---|---|---|---|
| GMI | - | 88.50 | 24.00 ± 2.86 | 44.22 ± 2.64 | 1712.33 | 27.52 |
| | MID | 83.82 | 18.02 ± 2.16 | 36.52 ± 2.30 | 1774.98 | 28.82 |
| | BiDO | 88.07 | 14.56 ± 1.44 | 34.02 ± 3.21 | 1809.92 | 33.63 |
| | NegLS | 84.68 | 8.94 ± 1.86 | 24.08 ± 1.97 | 1774.46 | 33.82 |
| | Trap-MID | 86.04 | **0.06 ± 0.15** | **0.52 ± 0.48** | **2471.01** | **180.65** |
| KED-MI | - | 88.50 | 76.88 ± 1.35 | 94.42 ± 0.91 | 1290.48 | 16.24 |
| | MID | 83.82 | 73.70 ± 2.26 | 92.34 ± 1.06 | 1295.08 | 19.27 |
| | BiDO | 88.07 | 63.48 ± 2.13 | 86.92 ± 1.76 | 1317.48 | 17.71 |
| | NegLS | 84.68 | 49.08 ± 2.77 | 77.96 ± 1.49 | 1413.04 | 25.70 |
| | Trap-MID | 86.04 | **0.42 ± 0.33** | **1.40 ± 0.49** | **2303.75** | **98.05** |
| LOMMA (GMI) | - | 88.50 | 84.24 ± 4.65 | 96.00 ± 2.43 | 1298.44 | 40.94 |
| | MID | 83.82 | 71.06 ± 5.35 | 90.00 ± 3.33 | 1381.04 | 40.52 |
| | BiDO | 88.07 | 83.76 ± 4.08 | 96.60 ± 2.43 | 1262.28 | 41.73 |
| | NegLS | 84.68 | 67.34 ± 5.42 | 87.40 ± 3.85 | 1302.95 | 38.68 |
| | Trap-MID | 86.04 | **45.36 ± 6.31** | **71.10 ± 5.28** | **1554.08** | **41.14** |
| LOMMA (KED-MI) | - | 88.50 | 89.82 ± 2.83 | 99.10 ± 1.18 | 1221.43 | 33.22 |
| | MID | 83.82 | 83.58 ± 3.02 | 97.10 ± 1.46 | 1221.44 | 22.76 |
| | BiDO | 88.07 | 86.28 ± 3.24 | 97.90 ± 1.50 | 1178.54 | 22.44 |
| | NegLS | 84.68 | 92.16 ± 2.62 | 99.10 ± 0.91 | 1155.51 | **34.28** |
| | Trap-MID | 86.04 | **62.24 ± 4.50** | **87.20 ± 2.68** | **1383.58** | 23.88 |
| PLG-MI | - | 88.50 | 99.62 ± 0.65 | 99.94 ± 0.24 | 1076.28 | 16.57 |
| | MID | 83.82 | 97.84 ± 1.10 | 99.58 ± 0.62 | 1051.97 | 13.29 |
| | BiDO | 88.07 | 99.68 ± 0.45 | 99.92 ± 0.28 | 991.42 | 19.46 |
| | NegLS | 84.68 | 97.30 ± 0.67 | 99.74 ± 0.17 | 1430.69 | **95.85** |
| | Trap-MID | 86.04 | **4.80 ± 1.99** | **12.86 ± 2.56** | **1971.50** | 47.46 |

Table 13: Defense comparison on ResNet-152 models.

| Attack | Defense | Acc ↑ | AA-1 ↓ | AA-5 ↓ | KNN Dist ↑ | FID ↑ |
|--------|---------|-------|--------|--------|------------|-------|
| GMI | - | 91.16 | 26.74 ± 2.80 | 48.94 ± 3.11 | 1681.49 | 28.20 |
| | MID | 91.29 | 34.72 ± 2.38 | 56.56 ± 2.26 | 1583.22 | 27.53 |
| | BiDO | 90.06 | 25.16 ± 2.04 | 46.64 ± 3.34 | 1738.49 | 32.66 |
| | NegLS | 83.93 | 8.20 ± 1.40 | 23.62 ± 3.19 | 1790.67 | 36.79 |
| | Trap-MID | 87.23 | **0.06 ± 0.14** | **0.52 ± 0.40** | **2475.04** | **165.47** |
| KED-MI | - | 91.16 | 74.50 ± 1.82 | 93.66 ± 1.70 | 1288.24 | 16.39 |
| | MID | 91.29 | 88.70 ± 1.04 | 98.08 ± 0.48 | 1127.35 | 17.14 |
| | BiDO | 90.06 | 67.46 ± 1.88 | 88.96 ± 1.43 | 1275.67 | 18.83 |
| | NegLS | 83.93 | 37.92 ± 2.77 | 67.18 ± 2.76 | 1481.29 | 32.18 |
| | Trap-MID | 87.23 | **0.58 ± 0.32** | **2.08 ± 0.63** | **2266.56** | **84.31** |
| LOMMA (GMI) | - | 91.16 | 83.02 ± 4.51 | 96.10 ± 2.64 | 1313.64 | 41.48 |
| | MID | 91.29 | 86.98 ± 3.93 | 97.30 ± 2.35 | 1219.14 | 42.10 |
| | BiDO | 90.06 | 65.96 ± 5.52 | 86.90 ± 4.51 | 1416.01 | 49.36 |
| | NegLS | 83.93 | 65.80 ± 6.16 | 86.00 ± 4.57 | 1322.77 | 37.84 |
| | Trap-MID | 87.23 | **45.70 ± 6.01** | **73.30 ± 4.47** | **1571.89** | **43.66** |
| LOMMA (KED-MI) | - | 91.16 | 90.22 ± 3.25 | 98.20 ± 1.20 | 1185.15 | 22.27 |
| | MID | 91.29 | 95.64 ± 2.19 | 99.90 ± 0.52 | 1068.42 | 23.47 |
| | BiDO | 90.06 | 70.86 ± 4.43 | 91.30 ± 2.65 | 1293.13 | 25.03 |
| | NegLS | 83.93 | 86.74 ± 3.43 | 98.10 ± 1.23 | 1208.54 | **36.68** |
| | Trap-MID | 87.23 | **69.86 ± 5.20** | **91.80 ± 2.78** | **1370.98** | 23.36 |
| PLG-MI | - | 91.16 | 99.34 ± 0.62 | 99.82 ± 0.37 | 1025.51 | 16.12 |
| | MID | 91.29 | 99.76 ± 0.48 | 99.84 ± 0.37 | 853.88 | 21.20 |
| | BiDO | 90.06 | 98.10 ± 0.97 | 99.62 ± 0.64 | 1042.27 | 28.48 |
| | NegLS | 83.93 | 94.82 ± 0.64 | 99.44 ± 0.30 | 1398.07 | **117.98** |
| | Trap-MID | 87.23 | **3.56 ± 1.38** | **11.16 ± 2.58** | **2016.62** | 73.84 |

Table 14: Defense comparison against PLG-MI with FFHQ dataset, using VGG-16 models.

| Defense | Acc ↑ | AA-1 ↓ | AA-5 ↓ | KNN Dist ↑ | FID ↑ |
|---------|-------|--------|--------|-----------|-------|
| - | 87.83 | 89.22 ± 1.60 | 97.72 ± 0.51 | 1278.05 | 18.36 |
| MID | 76.67 | 66.06 ± 4.17 | 87.54 ± 2.46 | 1371.81 | 15.75 |
| BiDO | 79.62 | 64.30 ± 3.48 | 86.00 ± 2.42 | 1416.98 | 18.37 |
| NegLS | 81.76 | 78.12 ± 1.13 | 92.60 ± 0.78 | 1506.23 | **109.67** |
| Trap-MID* | 81.62 | **0.86 ± 0.83** | **2.85 ± 1.31** | **2227.09** | 94.57 |

\* The mean and standard deviation of each evaluation metric are averaged across multiple runs.

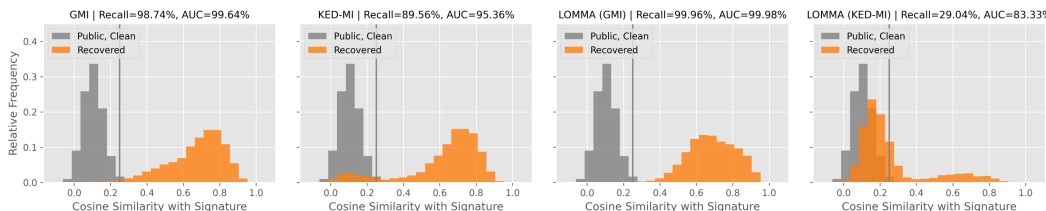

Figure 7: Illustration of trapdoor detection against different MI attacks.

## E.5 Distributional Shifts in Adversary's Auxiliary Dataset

In the previous experiments, we assumed that the adversary possesses a public dataset without distributional shifts from the private data. Typically, both auxiliary and private datasets were constructed from the CelebA dataset. However, in a practical scenario, the adversary might not know the distribution of private data, leading to potential distributional shifts between auxiliary and private datasets and making it harder to extract private data.

In this section, we used the FFHQ dataset [38] as the adversary's auxiliary dataset to demonstrate the scenario with slight distributional shifts, considering that PLG-MI can still provide an attack accuracy exceeding 89% on the unprotected model using this dataset [8]. As shown in Table 14, Trap-MID outperforms previous approaches under this scenario, achieving nearly 0% top-1 attack accuracy.

Notably, while methods using dependency regularization, such as MID and BiDO, are vulnerable to MI attacks without distributional shifts, they can protect privacy if auxiliary distribution differs from private data, with top-1 attack accuracy dropping from 89% to below 70%. On the other hand, although NegLS reduces the guidance signal for MI attacks by training an over-confident model with a discrete loss landscape, the logit-based max-margin loss in PLG-MI can prevent early saturation and surpass this defense mechanism. Consequently, while NegLS leads to unnatural reconstructions with a high FID, it remains vulnerable to PLG-MI, with attack accuracy exceeding 78%.

## E.6 Trapdoor Recovery Analysis against Different MI Attacks

Figure 7 presents the trapdoor recovery analysis of MI attacks other than PLG-MI. Similarly to PLG-MI, all these attacks, except LOMMA (KED-MI), reconstruct the trapdoor information from trapdoored models, with more than 89% recovered images reported as triggered images. Although our detection method has a low recall rate on LOMMA (KED-MI), the ROC AUC of 83.33% still suggests that the reconstructed images are likely to be injected with a trapdoor trigger. However, due to the limitations of backdoor attacks, the student models from KD enable LOMMA to extract trapdoor information and explore private distribution simultaneously, preventing it from being entirely misled by the trapdoored models. We leave the further adaptation for a robust trapdoor against KD to future works.

Table 15: Defense comparison against adaptive attacks, using VGG-16 models.

| Attack | Defense | Acc ↑ | AA-1 ↓ | AA-5 ↓ | KNN Dist ↑ | FID ↑ |
|--------|---------|-------|--------|--------|-----------|-------|
| PLG-MI | Trap-MID* | 81.37 | $6.23 \pm 1.70$ | $13.15 \pm 2.57$ | 2055.96 | 57.82 |
| PLG-MI+ | - | 87.83 | $98.78 \pm 1.05$ | $99.90 \pm 0.30$ | 1086.80 | 12.88 |
| | MID | 76.67 | $87.64 \pm 2.38$ | $97.38 \pm 1.29$ | 1208.60 | 15.41 |
| | BiDO | 79.62 | $89.64 \pm 2.36$ | $97.96 \pm 1.04$ | 1222.26 | 16.89 |
| | NegLS | 81.76 | $92.20 \pm 0.94$ | $98.38 \pm 0.55$ | **1395.18** | **77.32** |
| | Trap-MID* | 81.62 | $\mathbf{74.26 \pm 2.02}$ | $\mathbf{81.86 \pm 1.33}$ | 1356.54 | 24.03 |
| PLG-MI++ | Trap-MID* | 81.62 | $70.44 \pm 2.24$ | $78.14 \pm 1.29$ | 1399.84 | 27.17 |

\* The mean and standard deviation of each evaluation metric are averaged across multiple runs.

Table 16: Defense comparison against adaptive attacks with FFHQ dataset, using VGG-16 models.

| Attack | Defense | Acc ↑ | AA-1 ↓ | AA-5 ↓ | KNN Dist ↑ | FID ↑ |
|--------|---------|-------|--------|--------|-----------|-------|
| PLG-MI | Trap-MID* | 81.62 | $0.86 \pm 0.83$ | $2.85 \pm 1.31$ | 2227.09 | 94.57 |
| PLG-MI+ | - | 87.83 | $89.36 \pm 2.60$ | $96.96 \pm 1.11$ | 1267.65 | 15.32 |
| | MID | 76.67 | $63.92 \pm 3.56$ | $85.96 \pm 2.81$ | 1369.63 | 13.25 |
| | BiDO | 79.62 | $60.64 \pm 3.06$ | $83.22 \pm 1.61$ | 1430.52 | 17.29 |
| | NegLS | 81.76 | $85.52 \pm 0.81$ | $96.78 \pm 0.84$ | 1446.69 | **86.98** |
| | Trap-MID* | 81.62 | $\mathbf{21.85 \pm 2.16}$ | $\mathbf{31.32 \pm 1.98}$ | **1897.36** | 46.47 |
| PLG-MI++ | Trap-MID* | 81.62 | $31.03 \pm 2.53$ | $44.09 \pm 2.44$ | 1789.95 | 38.98 |

\* The mean and standard deviation of each evaluation metric are averaged across multiple runs.

### E.7 Adaptive Attacks without Trapdoor Signatures

This section investigates adaptive attacks when the adversary only knows the existence of trapdoors without access to trapdoor signatures. Here we modify the loss function in Equation 7 by excluding the regularization term of trapdoor signatures:

$$\mathcal{L}_{G+} = \mathcal{L}_G - \lambda_{\text{aux}} \mathbb{E}_{Y \sim p_{\text{aux}}(Y)} \big[ \mathbb{E}_{Z \sim p_G(Z)} [\cos(g_\theta(T_{\text{attack}}(G(Z, Y))), S_{\text{aux}, Y})] \big]. \tag{12}$$

This adaptive attack is denoted by PLG-MI+. Table 15 demonstrates its comparable attack performance with PLG-MI++, indicating that if the auxiliary dataset is close enough to the private distribution, guiding attacks by auxiliary signatures is sufficient to boost attacks. However, as Table 16 shows, trapdoor signatures are required to enhance the attack performance further when there are distributional shifts in auxiliary data. Overall, Trap-MID still provides better privacy preservation than existing defenses against adaptive attacks.

### E.8 Combining Trap-MID with NegLS

We further analyze whether Trap-MID can be combined with existing baselines to enhance performance. Table 17 presents the defense performance of combining Trap-MID with NegLS, using the same configurations outlined in Appendix C. Following the evaluation protocol in Section 4.1, the experiments are conducted across 5 runs. This hybrid defense slightly decreases accuracy but significantly improves defense effectiveness, even against KD-based attacks like LOMMA. For instance, it reduces LOMMA (KED-MI)'s attack accuracy to 42.47%, while Trap-MID or NegLS alone achieves only 61.25% or 77.67%. The reconstructed images can be found in Appendix F.

This suggests that Trap-MID stands as an orthogonal approach to existing defenses and can be integrated with them. Intuitively, NegLS focuses on reducing the leakage of guiding signals, making it more difficult for adversaries to extract private data. Consequently, this enhancement makes Trap-MID's shortcuts more appealing to attack algorithms, thereby bolstering privacy protection.

Table 17: Defense performance when combining Trap-MID with NegLS, using VGG-16 models.

| Defense | Acc ↑ | AA-1 ↓ | AA-5 ↓ | KNN Dist ↑ | FID ↑ |
|---|---|---|---|---|---|
| | | | GMI | | |
| Trap-MID | $81.37 \pm 1.04$ | $\mathbf{0.24 \pm 0.19}$ | $\mathbf{1.16 \pm 0.83}$ | $\mathbf{2411.39 \pm 80.80}$ | $\mathbf{153.73 \pm 62.84}$ |
| w/ NegLS | $77.10 \pm 0.89$ | $1.28 \pm 1.30$ | $4.43 \pm 3.99$ | $2244.13 \pm 89.50$ | $89.50 \pm 31.01$ |
| | | | KED-MI | | |
| Trap-MID | $81.37 \pm 1.04$ | $9.24 \pm 9.36$ | $19.24 \pm 18.65$ | $\mathbf{2056.00 \pm 311.59}$ | $\mathbf{87.39 \pm 66.40}$ |
| w/ NegLS | $77.10 \pm 0.89$ | $\mathbf{4.26 \pm 2.56}$ | $\mathbf{12.14 \pm 6.73}$ | $2004.05 \pm 164.76$ | $76.32 \pm 35.72$ |
| | | | LOMMA (GMI) | | |
| Trap-MID | $81.37 \pm 1.04$ | $41.63 \pm 2.28$ | $68.24 \pm 2.60$ | $1569.92 \pm 19.74$ | $39.29 \pm 2.14$ |
| w/ NegLS | $77.10 \pm 0.89$ | $\mathbf{22.80 \pm 3.12}$ | $\mathbf{46.88 \pm 4.87}$ | $\mathbf{1710.01 \pm 48.15}$ | $\mathbf{47.44 \pm 4.17}$ |
| | | | LOMMA (KED-MI) | | |
| Trap-MID | $81.37 \pm 1.04$ | $61.25 \pm 5.71$ | $85.76 \pm 3.73$ | $1404.77 \pm 40.25$ | $24.19 \pm 2.21$ |
| w/ NegLS | $77.10 \pm 0.89$ | $\mathbf{42.47 \pm 8.97}$ | $\mathbf{70.64 \pm 9.16}$ | $\mathbf{1521.82 \pm 73.60}$ | $\mathbf{37.22 \pm 1.56}$ |
| | | | PLG-MI | | |
| Trap-MID | $81.37 \pm 1.04$ | $6.23 \pm 5.60$ | $13.15 \pm 10.30$ | $2055.96 \pm 147.67$ | $57.82 \pm 23.41$ |
| w/ NegLS | $77.10 \pm 0.89$ | $\mathbf{0.66 \pm 0.68}$ | $\mathbf{1.96 \pm 1.30}$ | $\mathbf{2344.62 \pm 58.48}$ | $\mathbf{93.66 \pm 22.14}$ |

Table 18: Defense comparison against BREP-MI, using untargeted attacks to recover 300 identities.

| Defense | Acc ↑ | Number of Initial Iterations ↑ | AA-1 ↓ |
|---|---|---|---|
| - | 87.83 | 2 | 65.00 |
| MID | 76.67 | 2 | 46.33 |
| BiDO | 79.62 | 3 | 39.00 |
| NegLS | 81.76 | 3 | 52.00 |
| Trap-MID | 81.62 | $\mathbf{171}$ | $\mathbf{0.00}$ |

Another promising direction for future research is to combine multiple defense strategies to improve overall performance and robustness against specific adaptive attacks.

### E.9 Defense Performance against Label-Only Attacks

In previous experiments, we evaluated defense methods against white-box MI attacks, which pose a greater privacy threat. However, in practical scenarios, adversaries may only have access to model predictions or output labels, conducting black-box or label-only attacks on victim models.

This section investigates the defense performance of different approaches against BREP-MI [11], a label-only attack. BREP-MI starts by randomly sampling the generator's latent until the victim model classifies the generated image as the target class. It then estimates the predicted labels over a sphere in latent space to iteratively adjust the image away from the model's decision boundary. We use the victim models trained on the CelebA dataset and the generator from GMI to analyze performance against BREP-MI.

Under targeted attack settings, BREP-MI fails to initialize latents for all 1,000 identities in a reasonable time when facing Trap-MID, sampling latents for only 942 identities after 820,000 iterations. In contrast, it only takes 553 iterations against unprotected models.

We also conducted untargeted attacks to recover 300 identities. As shown in Table 18, Trap-MID significantly increased the number of initial iterations required and reduced BREP-MI's attack accuracy to 0%, demonstrating its effective privacy protection against various types of MI attacks.

Table 19: Defense performance against PPA.

| Architecture | Defense | Acc ↑ | AA-1 ↓ | AA-5 ↓ | $\delta_{\text{face}}$ ↑ | $\delta_{\text{eval}}$ ↑ | FID ↑ |
|---|---|---|---|---|---|---|---|
| ResNeSt-101 | -* | 87.35 | 82.96 | 95.44 | 0.7506 | 299.73 | 44.04 |
| | Trap-MID | 88.48 | 74.47 | 89.61 | 0.8188 | 319.48 | 42.36 |
| | + Fine-tuned | 83.22 | **15.58** | **26.95** | **1.3162** | **471.61** | **64.34** |
| ResNet-152 | -* | 86.78 | 80.61 | 94.58 | 0.7362 | 312.58 | 40.43 |
| | NegLS[†] | 83.59 | 26.41 | 49.96 | 1.0420 | 441.67 | 61.30 |
| | Trap-MID | 88.02 | 65.93 | 83.17 | 0.8606 | 344.45 | 41.61 |
| | + Fine-tuned | 83.82 | **0.25** | **0.90** | **1.5807** | **559.08** | **66.54** |
| DenseNet-169 | -* | 85.39 | 73.14 | 90.51 | 0.7635 | 312.32 | 43.24 |
| | Trap-MID | 89.98 | 55.53 | 71.25 | 0.9571 | 358.67 | 49.37 |
| | + Fine-tuned | 86.82 | **1.79** | **4.15** | **1.5646** | **552.48** | **74.47** |

\* Reported in PPA's paper [6].
[†] Reported in NegLS's paper [16].

Table 20: Additional evaluation results of the defense performance against PPA.

| Architecture | Defense | Precision ↓ | Recall ↓ | Density ↓ | Coverage ↓ |
|---|---|---|---|---|---|
| ResNeSt-101 | -* | 0.2650 | 0.0136 | 0.8547 | 0.3624 |
| | Trap-MID | 0.3195 | **0.0019** | 0.9659 | 0.4329 |
| | + Fine-tuned | **0.1320** | 0.0224 | **0.3999** | **0.2806** |
| ResNet-152 | -* | 0.3231 | 0.0269 | 0.7984 | 0.2805 |
| | Trap-MID | 0.1873 | 0.0608 | 0.7479 | 0.4556 |
| | + Fine-tuned | **0.0680** | **0.0110** | **0.2931** | **0.1913** |
| DenseNet-169 | -* | 0.2049 | 0.0495 | 0.6811 | 0.3866 |
| | Trap-MID | 0.2976 | **0.0127** | 0.9781 | 0.5286 |
| | + Fine-tuned | **0.0723** | 0.1025 | **0.2565** | **0.1368** |

\* Reported in PPA's paper [6].

## E.10 Defense Performance Under High-Resolution Scenario

This section evaluates Trap-MID's defense performance on modern architectures and in high-resolution scenarios. Specifically, we conducted the experiments introduced by PPA [6]. The target models are ResNeSt-101 [46], ResNet-152 [33], and DenseNet-169 [47] trained on a high-quality version of the CelebA dataset, with the images cropped, aligned, and resized to 224x224 using HD CelebA Cropper.[3] We adopted PPA to perform MI attacks using a StyleGAN2 generator [21] pre-trained on the FFHQ dataset [38]. The generator outputs 1024x1024 images, which are then center-cropped to 800x800 and resized to fit the model's input resolution.

We adopt the same InceptionV3 model used in PPA's official setup as the evaluation model. It was trained on the same training dataset as the target models, with an input size of 299x299. The evaluation model achieves 93.28% accuracy on the testing dataset. In addition to attack accuracy, KNN distance in the evaluation model's feature space ($\delta_{\text{eval}}$), and FID, PPA also employs additional metrics, such as KNN Distance in the FaceNet [43] feature space ($\delta_{\text{face}}$), improved precision and recall [44], and density and coverage [45] metrics. Details of these metrics are provided in Appendix E.3.

We follow the same training settings as [6]. In addition to the Trap-MID configurations detailed in Appendix C.4, we also fine-tuned the blend ratio $\alpha$ and trapdoor loss weight $\beta$ according to the observations in Appendix D.3 and D.4. Since we found that Trap-MID models generally achieve better accuracy than the unprotected models, we set $\beta = 0.5$ and selected the smallest $\alpha$ for each model to effectively distinguish triggered samples from benign ones. Specifically, the blend ratios are 0.005, 0.007, and 0.01 for the ResNeSt-101, ResNet-152, and DenseNet-169 models, respectively.

---

[3]Code available at `https://github.com/LynnHo/HD-CelebA-Cropper`.

Table 19 and Table 20 present Trap-MID's defense performance against PPA, compared with attack results reported in previous works. Although Trap-MID does not fully mitigate PPA with default settings, it preserves privacy to a certain degree without sacrificing accuracy. Furthermore, selecting proper hyper-parameters significantly enhances defense performance, reducing attack accuracy on ResNet-152 and DenseNet-169 to below 2%, and that on ResNeSt-101 to 15.58%. This demonstrates that while hyper-parameter tuning is essential for optimal defense, Trap-MID provides effective privacy protection across various datasets and architectures,

## F Additional Visualization

Figure 8 shows the reconstructed images with the hybrid approach discussed in Appendix E.8. Figure 9 illustrates additional recovered images from PLG-MI, where the recoveries of Identity 4 from Trap-MID exhibit different hair colors from the private data. Such variation does not appear in other defenses, demonstrating Trap-MID's effectiveness in protecting private information. In addition, Figure 10, Figure 11, Figure 12, and Figure 13 display sample reconstructed images from GMI, KED-MI, and LOMMA attacks.

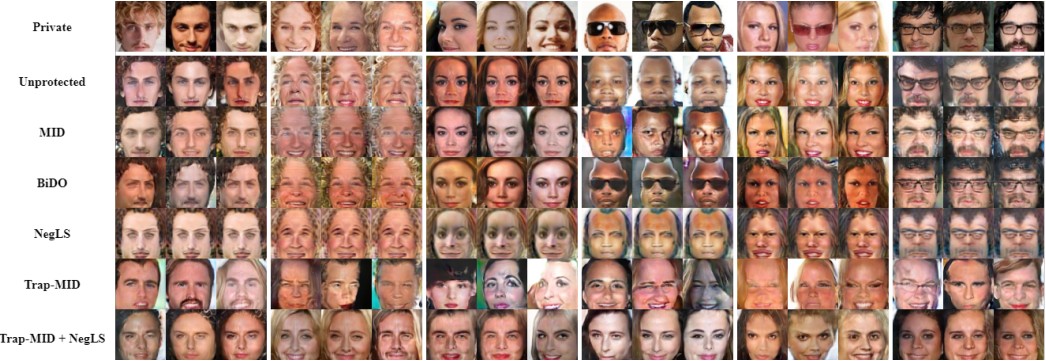

Figure 8: Reconstructed images from PLG-MI, including Trap-MID + NegLS.

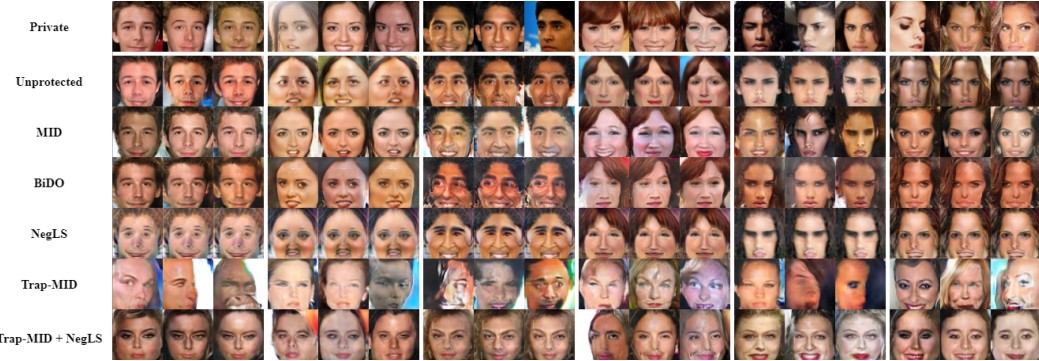

Figure 9: Additional reconstructed images from PLG-MI.

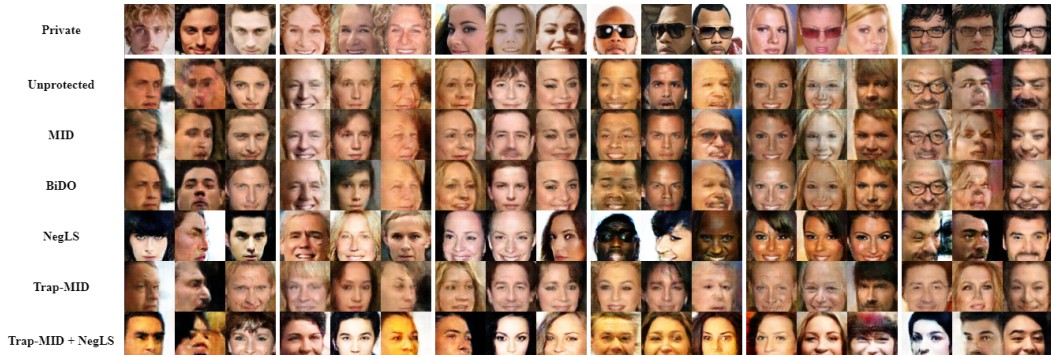

Figure 10: Reconstructed images from GMI.

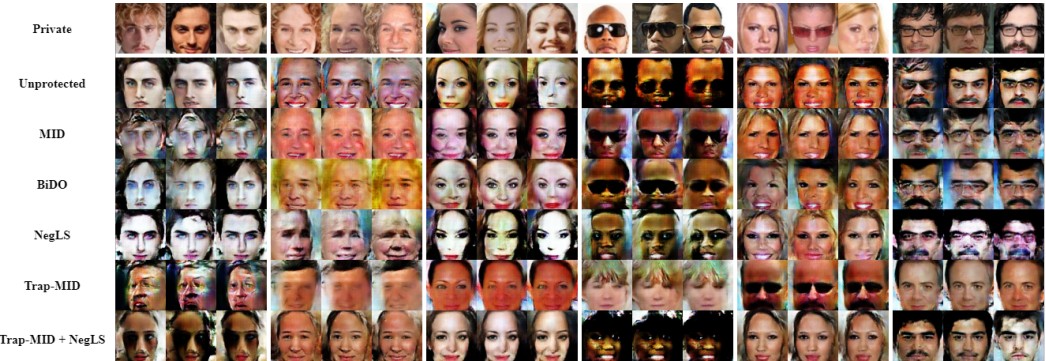

Figure 11: Reconstructed images from KED-MI.

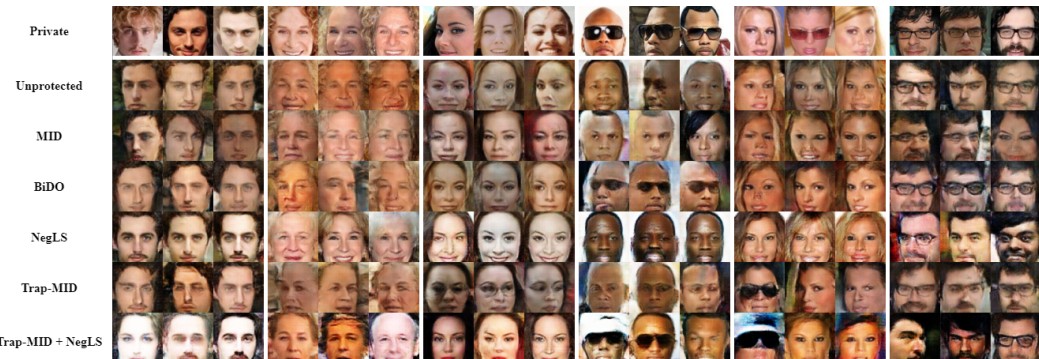

Figure 12: Reconstructed images from LOMMA (GMI).

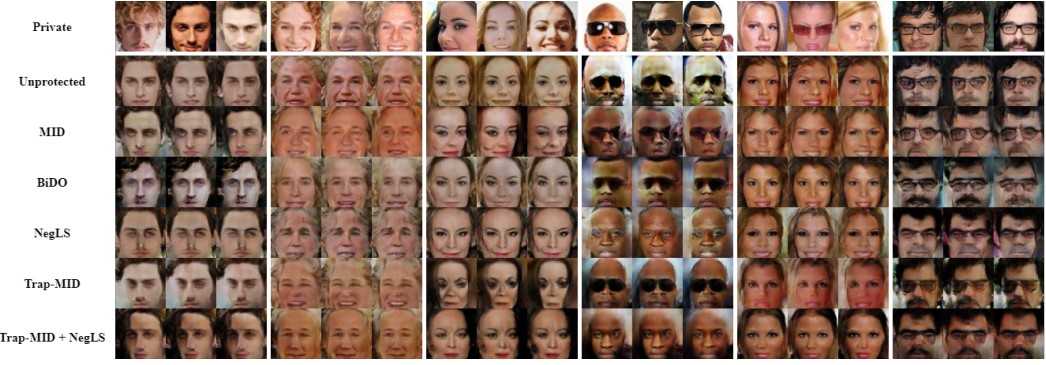

Figure 13: Reconstructed images from LOMMA (KED-MI).

