# OpenReview forum: "Trap-MID: Trapdoor-based Defense against Model Inversion Attacks"
_NeurIPS.cc/2024/Conference — NeurIPS 2024 poster_

### Official Review · Reviewer_bcsy · 2024-06-21

**Soundness:** 3
**Presentation:** 3
**Contribution:** 3
**Rating:** 6
**Confidence:** 5

**Summary:**

The paper proposes Trap-MID, a novel defense method against model inversion attacks, drawing inspiration from backdoor attacks and shortcut-based defenses against adversarial examples. The core idea involves adding poisoned data samples to the target model's training data. This data poisoning is based on the Blended backdoor attack, which calculates a linear combination of the training sample and a noise trigger pattern. To reduce the visibility of the trigger, a discriminator is trained in parallel, akin to a GAN training setup. This discriminator is then used to align the trigger patterns with the distribution of the clean training samples. The proposed defense method is empirically evaluated against common white-box attacks (GMI, KED-MI, LOMMA, PLG-MI) and defense methods (MID, BiDO, NegLS) using the standard 64x64 CelebA training samples.

**Strengths:**

- The paper proposes an intriguing direction for model inversion defenses by utilizing backdoor attacks to induce shortcuts in the model. These shortcuts are then exploited by the attack, resulting in misleading attack outcomes. This approach is both clever and conceptually straightforward, which is advantageous.

- The paper is well-written, with all components and sections clearly described. The experimental evaluation is also well-defined, adhering to the standard procedures in model inversion literature.

- The results presented in the evaluation section are convincing. The experiments demonstrate the method's favorable privacy-utility trade-off and its effective defense capabilities. Additionally, the paper investigates adaptive attacks, a critical aspect of assessing a method's effectiveness. Although the adaptive attack still achieves good results against the proposed defense, this does not diminish the contribution's value. The Appendix further provides an extensive ablation and sensitivity analysis, essential for understanding the impact of hyperparameter selection and individual design choices.

**Weaknesses:**

- Although the evaluation is conducted on four attack algorithms, two important aspects are missing in my opinion. First, the proposed defense should also be tested against black-box/label-only attacks. In particular, testing the defense method against the label-only BREP-MI [1] attack would be very interesting since this method aims to reconstruct samples by maximizing their distance to the decision boundaries. Given that the proposed Trap-MID leverages a model's ability to exploit shortcuts in its predictions, I am curious if the poisoned shortcut samples are differently placed in a model's embedding space. It might be the case that these samples are actually closer to the decision boundary compared to clean training samples. If this is true, the defense might fail. So even if white-box attacks are usually considered stronger, running some label-only attacks could add an intriguing perspective to the paper.

-  Similarly, the defense is only evaluated in a low-resolution setting. However, it is important to also investigate a high-resolution setting, which can be considered more practical. In this context, the method should also be tested against PPA [2] to provide a comprehensive evaluation and see if the promised defense effect holds in this setting. I want to emphasize that I am not requesting additional experiments merely as a form of criticism, but because investigating these two additional attacks would add significant value to the paper, helping to support the claims.

- The evaluation heavily relies on attack accuracy and KNN Distance, both computed on an evaluation model trained on the target dataset. However, I think these metrics are limited as they tell us little about the actual visual similarity between reconstructed results and the training samples. For example, in Fig. 2, the images reconstructed from the NegLS model look unrealistic and reveal only limited information about the target identity. Still, the Attack Accuracy in Tab. 1 is high for PLG-MI and the KNN-Dist is low. Both metrics seem very susceptible to adversarial noise and misguided attack results. Similarly, the FID score only assesses the image quality of the reconstructed samples and compares it to the training data. But this metric has no meaning regarding privacy leakage: generating samples of the same style as the target training data leads to a low FID, even if the images reveal no private attributes. Conversely, images following a substantially different distribution can still reveal private features, even if the FID score is high. Additional metrics are therefore necessary to assess the actual effectiveness. From literature, one could use a model like CLIP or for the face recognition setting, a FaceNet model to assess the identity similarity between attack results and training samples (see, e.g., [2]). Another option would be the knowledge extraction score introduced in [4], which measures the information content included in the attack results.

- While I appreciate that the paper includes a theoretical analysis of the method and its effectiveness, I have some doubts about the formal proof. Particularly, I do not think that the KL divergence is a valid measure here to assess the trapdoor visibility. The problem is that the KL divergence compares two distributions. However, similar distributions might not necessarily mean that the triggers are invisible, and vice versa. For example, one could use triggers that are clearly visible but follow the clean data distribution. For example, using physical triggers instead of noise, but also noise patterns could be designed to follow the clean data distribution. This limits the expressiveness of the theorem since it relies on the KL divergence. In my view, the analysis should use a sample-wise measurement that compares a clean sample directly with its poisoned counterpart to provide a reliable measurement of trigger visibility.

- The work of [6] investigates a similar direction (using surrogate samples to misguide the attack). While their approach is partly limited in the number of classes to defend, I think the approach should at least be discussed in the related work section since the underlying idea of providing false guidance for the attack is conceptually similar.


Small Remarks:
- L118: "Given access to the target classifier f, the adversary aims to recover the private data of class y by estimating the posterior distribution p(X|y)..." -> While this definition is not wrong, I think model inversion attacks do not necessarily require the adversary to recover the whole posterior distribution; reconstructing a single sample can be enough for a serious privacy breach. While there exist attacks directly aiming to reconstruct the posterior, e.g., VMI [3], this might not be true for all attacks.
- There is another recent defense that might be included in the related work section [5]. The paper was very recently accepted at CVPR and only became available close to the NeurIPS deadline. So, I do not expect the paper to include this work in its evaluation. Just wanted to highlight this related work.
- L517: I think the URL is the wrong one here, as it leads to the KED paper.
- L525: If I have not missed it, the paper never actually defines the evaluation model architecture. Defining this architecture once helps the reproducibility of the paper.

-> I encourage the authors to participate in the rebuttal and will increase my scores if the weaknesses mentioned are addressed adequately.


References:
[1] Kahla et al., Label-Only Model Inversion Attacks via Boundary Repulsion. CVPR 2022
[2] Struppek et al., Plug & Play Attacks: Towards Robust and Flexible Model Inversion Attacks. ICML 2022
[3] Wang et al., Variational model inversion attacks. NeurIPS 2021
[4] Struppek et al., Be Careful What You Smooth For: Label Smoothing Can Be a Privacy Shield but Also a Catalyst for Model Inversion Attacks. ICLR 2024
[5] Ho et al., Model Inversion Robustness: Can Transfer Learning Help? CVPR 2024
[6] Chen et al. Data-Centric Defense: Shaping Loss Landscape with Augmentations to Counter Model Inversion. ICML Workshop 2023

**Questions:**

- Can inference-time mitigation strategies from the backdoor or shortcut literature be employed to eliminate the model's shortcut behavior, thereby enabling stronger attacks?
- Table 6 in the appendix: Why does the attack performance increase for other attacks besides PLG-MI when the defense uses the trapdoor loss and a discriminator loss compared to fixed triggers? Is there any intuition behind these results?

**Limitations:**

Section A.2. provides a comprehensive limitations section.

---

> ### Author Rebuttal · Authors · 2024-08-07
>
> Thank you for your insightful review and valuable feedback. We address the specific weaknesses and questions raised in your comments below:
>
> **W1: The defense should be tested against black-box/label-only attacks**
>
> The experiments show that BREP-MI, a label-only attack, is ineffective on Trap-MID, requiring over 820,000 iterations to initialize latents for all identities and getting 0% attack accuracy in untargeted attacks recovering 300 targets, which indicates Trap-MID's efficacy. Detailed results are available in Table 1 of the attached file in Author Rebuttal. We will add this analysis to our paper.
>
> **W2: It is important to investigate a high-resolution setting**
>
> We add the experiments against PPA for a high-resolution setting. Due to time constraints, we modified the attack to optimize 20 samples and select 5 in the final stage (PPA originally optimized 200 samples and selected 50).
>
> The following table shows the defense performance on DenseNet-169. Although Trap-MID doesn't fully mitigate PPA, it preserves privacy to a certain degree without an accuracy drop. Besides, increasing trapdoor loss weight or combining Trap-MID with NegLS can improve the defense further. We discuss more about the defense combination in Q1.
>
> |Defense|Acc ↑|AA-1 ↓|AA-5 ↓|$\delta_{face}$ ↑|$\delta_{eval}$ ↑|FID ↑|
> |-|-|-|-|-|-|-|
> |Trap-MID|89.98|59.48|72.22|0.9468|182.90|49.64|
> |$\beta$=0.5|83.39|11.78|19.24|1.3985|269.99|63.04|
> |w/ NegLS|83.56|**1.60**|**4.90**|**1.4744**|**279.68**|**79.37**|
>
> Results including other models can be found in Table 2 of the attached file. We regret that we cannot conduct hyper-parameter tuning on other models due to time constraints. We will add this analysis to our paper.
>
> **W3: The evaluation metrics are limited**
>
> We acknowledge that each metric has limitations, yet attack accuracy/KNN distance and FID can complement each other. The formers estimate the extracted attributes, while the latter measures the naturalness and styling similarity. However, we agree that a universal metric would be valuable for a straightforward comparison.
>
> Our PPA experiments at W2 include additional metrics such as FaceNet's feature distance used by PPA and improved precision, recall, density, and coverage used by VMI. We also use them to evaluate PLG-MI's results for a more comprehensive analysis. The results are available in Tables 2 and 3 of the attached file. We will add this analysis to our paper.
>
> **W4: The analysis should use a sample-wise measurement to provide a reliable measurement of trigger visibility**
>
> Thank you for highlighting this concern. The "trapdoor visibility" in Definition 2 might be misleading. We use KL divergence to ensure that triggered data is natural enough to be generated in attacks. Making them similar to the original images is one method to achieve this objective. If triggered data is similar enough to its counterpart, we have
>
> $$\text{If } \forall x \in X,\\; \log{p(x)} - \log{p(\Pi(x))} \le \epsilon,\text{ then } D_{KL}(p(X)||p(\Pi(X)))  \le \epsilon,$$
>
> which fulfills Definition 2.
>
> We believe this theoretical analysis suggests the potential of various trigger designs. For instance, if anyone in a green shirt is classified as identity 1, the attacks could be misled into manipulating shirt colors.
>
> **W5: Missing related work**
>
> Thank you for mentioning these works. We will add them to Related Work.
>
> **W6: The definition of objective might not be true for all attacks**
>
> This definition doesn't mean that attacks aim to recover the entire posterior distribution. Instead, the attack relies on such distribution. Most MI attacks guide optimization by identity and prior loss, maximizing $p(y|X)$ and $p(X)$. According to Bayes' theorem, given target label $y$, the attack process can then be viewed as maximizing the posterior probability $p(X|y)$.
>
> We will refine the description here to make it clearer.
>
> **W7: The link to NegLS's implementation**
>
> Since NegLS hadn't released their code when we conducted experiments, we adapted KED-MI's code to implement NegLS by ourselves. We will include the link to our source code after releasing it.
>
> **W8: Missing evaluation model's architecture**
>
> Following GMI, the evaluation classifier is Face.evoLVe with an input size of 224x224. We will add this information to our paper.
>
> **Q1: Can inference-time mitigation strategies from the backdoor/shortcut literature eliminate the model's shortcut behavior?**
>
> If such mitigations are differentiable, one method is to apply them before feeding synthetic data into the victim model. Our work is a first step in trapdoor-based defense. Future work could explore the impact of recent backdoor/shortcut techniques on both attacks and defenses.
>
> Besides, we found that combining Trap-MID and NegLS further improves defense, suggesting that Trap-MID can complement existing methods. Intuitively, NegLS makes it harder to extract private data and therefore makes trapdoors more attractive. A future direction explores hybrid defense to counter specific adaptive attacks.
>
> |Attack|Defense|Accuracy ↑|AA-1 ↓|AA-5 ↓|KNN Dist ↑|FID ↑|
> |-|-|-|-|-|-|-|
> |LOMMA (KED-MI)|Trap-MID|81.37|61.25|85.76|1404.77|24.19|
> ||w/ NegLS|77.10|**42.47**|**70.64**|**1521.82**|**37.22**|
>
> Results against other attacks can be found in Table 4 of the attached file. We will add this analysis to our paper.
>
> **Q2: Why does the attack performance increase for other attacks besides PLG-MI when using trapdoor loss and discriminator loss?**
>
> This may come from the capacities of GAN in attacks.
>
> Trapdoor loss search for easier triggers for the target model to retain utility. However, these crafted triggers may be harder to generate, leading to a lower defense performance against weaker generators.
>
> Discriminator loss encourages invisible triggers, which is essential to deceive stronger discriminators like PLG-MI. However, generating invisible triggers requires fine-grain adjustment, making the weaker generators less likely to be misled.

---

> > ### Comment · Reviewer_bcsy · 2024-08-09
> >
> > I thank the authors for providing additional details and experiments. Many of my remarks and weaknesses have been addresses. Yet, some weaknesses and questions remain:
> >
> > 1.) I still believe the evaluation metrics are limited, using only the evaluation model + the FID score (see my initial review). I also do not agree that the FID score actually complements the other two metrics, since it has no clear implication on privacy leakage. Why not using something like FaceNet also for the 64x64 experiments?
> > 2.) "Since NegLS hadn't released their code..." -> I just checked, the authors refer to the code in their paper and the corresponding Github repo also seems to provide configurations for training models with negative LS. Yet, it seems like the authors implemented the defense by themselves, which should also be fine.
> > 3.) The impact of Trap-MID in defending in the high-resolution setting against PPA seems limited. I agree that it offers a nice improvement over the negative LS defense, but on its own the defense seems rather weak.
> > 4.) I also agree with Reviewer P6SC that the runtime is an important aspect. While the runtime is still somewhat reasonable, it limits the approach to some extent.
> > 5.) Regarding the evaluation model, is the input size of the Face.evolve model really 224x224? It seems somewhat strange that the evaluation model for the 64x64 setting requires an upscaling of factor 4 on the attack samples.
> >
> > Overall, after reading all reviews and the corresponding rebuttals, I decided to keep my initial positive score.

---

> ### Author Response · Authors · 2024-08-10
>
> Thank you for your thoughtful questions and feedback. We address them as follows:
>
> **Q1: Why not use something like FaceNet also for the 64x64 experiments?**
>
> The FaceNet evaluation of PLG-MI's 64x64 experiments can be found in Table 3 of our attached file at Author Rebuttal, which includes the following additional metrics:
>
> 1. **$\delta_{face}$**: The **FaceNet feature distance** between each recovered image and the nearest private data. A higher value indicates less similarity to the private data.
> 2. **Improved Precision**: Measures whether each recovered image lies within the estimated manifold of private data in the **InceptionV3 feature space**. A lower value indicates less similarity to the private data.
> 3. **Improved Recall**: Evaluates whether each private image lies within the manifold of recovered data in the **InceptionV3 feature space**. A lower value suggests that private data is less likely to be reproduced by the generator.
> 4. **Density**: Quantifies how many private-sample neighborhood spheres contain each recovered image in the **InceptionV3 feature space**. A lower value indicates less similarity to the private data.
> 5. **Coverage**: Assesses how many private samples have a neighborhood sphere containing at least one recovered image in the **InceptionV3 feature space**. A lower value suggests that private data is less likely to be reproduced by the generator.
>
> As shown in the results below, Trap-MID outperforms existing defenses in FaceNet distance and ranks second to NegLS in most other metrics.
>
> |Defense|Acc ↑|$\delta_{face}$ ↑|Precision ↓|Recall ↓|Density ↓|Coverage ↓|
> |-|-|-|-|-|-|-|
> |-|87.83|0.6110|19.39|13.17|0.0893|0.1498|
> |MID|76.67|0.6410|21.25|33.96|0.0913|0.1734|
> |BiDO|79.62|0.7058|20.17|10.17|0.0807|0.1367|
> |NegLS|81.76|0.7587|**3.80**|**0.00**|**0.0244**|**0.0189**|
> |Trap-MID|81.62|**1.3845**|9.56|71.63|0.0328|0.0728|
>
> We notice that Trap-MID's ability to classify arbitrary images with injected triggers as corresponding classes results in more diverse recovered images, leading to a broader manifold and higher recall. Besides, all metrics except FaceNet distance utilize the same InceptionV3 model as FID, making NegLS excel in these metrics with its less natural recovered images.
>
> **Q2: It seems like the authors implemented the defense by themselves, which should also be fine**
>
> Yes, the authors of NegLS released their code after we implemented it. We have verified that our implementation adheres to the original training algorithm, and the detailed configurations can be found in Appendix C.6 in our paper.
>
> **Q3: The impact of Trap-MID in defending in the high-resolution setting against PPA seems limited**
>
> It is worth noting that with proper configuration, Trap-MID can still perform effectively against PPA. For example, as shown in Table 2 of our attached file, Trap-MID reduces PPA's attack accuracy to 11.78% on DenseNet-169 when $\beta=0.5$. However, we acknowledge that tuning hyper-parameters is necessary for optimal defense across different datasets and architectures.
>
> Besides, we noticed that the success of hybrid defense like Trap-MID + NegLS is not merely owed to either of them but lies in their orthogonal strategies that (1) make it harder to extract private data and (2) provide misleading shortcuts. For instance, as shown in Table 2 of our paper and Table 4 of the attached file, this combination reduces LOMMA (KED-MI)'s attack accuracy to 42.47%, whereas either method alone achieves only 61.25% or 77.67%.
>
> **Q4: While the runtime is still somewhat reasonable, it limits the approach to some extent**
>
> We agree that computational efficiency is an important consideration, despite the effectiveness of Trap-MID. We will add this discussion to our Future Work section.
>
> **Q5: Regarding the evaluation model, is the input size of the Face.evolve model really 224x224?**
>
> We apologize for the mistake. The input size of the evaluation model is actually 112x112. The 64x64 images are indeed resized to fit the model's input size.

---

> > ### Comment · Reviewer_bcsy · 2024-08-10
> >
> > Thank you for addressing my additional questions. Regarding Q1, that was my mistake; I misunderstood the experimental setting in the table when I first reviewed it. I have no further questions and will maintain my previous score.

---

> > > ### Author Response · Authors · 2024-08-11
> > >
> > > We sincerely appreciate your positive feedback and valuable comments! Your insights and suggestions are instrumental in refining our work.

---

### Official Review · Reviewer_BGpu · 2024-07-01

**Soundness:** 2
**Presentation:** 3
**Contribution:** 3
**Rating:** 5
**Confidence:** 5

**Summary:**

The paper introduces a backdoor-based MI attack called Trap-MID. In this method, a trapdoor is integrated into the model to predict a specific label when the input is injected with the corresponding trigger. Consequently, this trapdoor information acts as a "shortcut" for MI attacks, causing them to extract trapdoor triggers instead of private data.

**Strengths:**

The paper provides the new insights about Model Inversion from Backdoor injection perspective.

The idea is intuitive and theoretical analysis is provided.

**Weaknesses:**

In line 137, “MI attacks leverage a discriminator to approximate generic prior and ensure natural outcomes” This statement might not generalize to all SOTA MI attacks those does not utilise a GAN discriminator such as PPA [1]

The evaluation is based on low-resolution (64x64) MI attacks, which are not very practical for modern models. I suggest authors to evaluate the effectiveness of Trap-MID and compare with baseline defenses on high-resolution setup such as PPA [1] or MIRROR [2]

The main results are on VGG-16, which is a quite out-dated architecture. Despite there are other results on Face.evoLVE and IR-152 in the Appx. I strongly encourage authors to evaluate on other architectures, such as in PPA [1]

The visualisation results are not very convincing to me. For example, in Fig. 2, most of the reconstructed images are very different from private data despite some defence baselines still have a high attack accuracy. I suggest reviewers to provide more visualisation for reference.

I understand that the visualisation could be subjective. I suggest the authors to conduct a comprehensive study to further confirm the effectiveness of Trap-MI when comparing with other defence baselines.

[1] Struppek, Lukas, et al. "Plug & play attacks: Towards robust and flexible model inversion attacks." ICML-2022.

[2] An, Shengwei et al. MIRROR: Model Inversion for Deep Learning Network with High Fidelity. Proceedings of the 29th Network and Distributed System Security Symposium.

**Questions:**

I have a question about the sensitivity of Trap-MI to hyper-parameters. We know that BiDO sensitive to hyper-parameters as it need to optimise two hyper-parameters. From my understanding, Trap-MI also introduces two hyper-parameters. I wonder how sensitive is the Trap-MI to these two hyper-parameters?

---

> ### Author Rebuttal · Authors · 2024-08-07
>
> Thank you for your thoughtful review and helpful comments. We address the specific weaknesses and questions raised in your review below:
>
> **W1: “MI attacks leverage a discriminator to approximate generic prior and ensure natural outcomes” This statement might not generalize to MI attacks that don't utilise GAN discriminator, such as PPA**
>
> We appreciate your feedback. This statement refers to the challenge of misleading MI attacks using trapdoors. Since MI attacks often rely on GAN to produce natural outcomes without explicit constraints like $l_2$ or $l_\infty$ distance in adversarial attacks, it becomes harder to design effective triggers.
>
> PPA uses a pre-trained StyleGAN2 generator and optimizes latents during attacks. Although the discriminator isn't involved in the process, it guides the generator to produce realistic images during GAN training, which indirectly ensures natural outcomes. We will refine this discussion to make it clearer.
>
> **W2: The evaluation is based on low-resolution (64x64) MI attacks, which are not very practical for modern models. The main results are on VGG-16, which is quite outdated.**
>
> To address this, we have added experiments against PPA to demonstrate a high-resolution (224x224) scenario with modern models. Due to time and computational constraints, we modified the attack to optimize 20 samples and select 5 in the final stage (PPA originally optimized 200 samples and selected 50, which would take about 10 days on our machine).
>
> The following table shows the defense performance on DenseNet-169. Although Trap-MID doesn't fully mitigate PPA with default settings, it preserves privacy to a certain degree without an accuracy drop. Besides, increasing trapdoor loss weight further improves defense performance, and combining Trap-MID with NegLS (keeping $\beta$=0.2) even reduces attack accuracy to 2%. This result demonstrates Trap-MID's effectiveness in privacy protection under different scenarios. We discuss more about the defense combination in Q3 of Author Rebuttal.
>
> |Defense|Acc ↑|AA-1 ↓|AA-5 ↓|$\delta_{face}$ ↑|$\delta_{eval}$ ↑|FID ↑|
> |-|-|-|-|-|-|-|
> |-|87.08|92.00|98.28|0.7240|140.93|37.76|
> |Trap-MID|89.98|59.48|72.22|0.9468|182.90|49.64|
> |$\beta$=0.5|83.39|11.78|19.24|1.3985|269.99|63.04|
> |w/ NegLS|83.56|**1.60**|**4.90**|**1.4744**|**279.68**|**79.37**|
>
> Detailed results, including other architectures, can be found in Table 2 of the attached file in Author Rebuttal We regret that we cannot conduct hyper-parameter tuning on other models due to time constraints. We will add this analysis to our paper.
>
> **W3: The visualization results are not very convincing to me. I suggest the authors provide more visualization for reference and conduct a comprehensive study to further confirm the effectiveness of Trap-MI.**
>
> We acknowledge that visualization can be subjective due to the ambiguous definition of "class representative." However, in Figure 2 of our paper, we found that recovered images from Trap-MID show more distinct attributes from private data. For example, Identity 1 images have different skin tones, Identity 2 and 5 images have different hairstyles, and Identity 4 images show different genders. We also provide more recovered samples in Figure 1 of the attached file in Author Rebuttal, in which Identity 4 images from Trap-MID have different hair colors from private data compared to other defenses.
>
> For the quantitative analysis, our PPA experiments in W2 include additional evaluation metrics, such as FaceNet's feature distance used in [1], as well as improved precision, recall, density, and coverage used in [2]. We also conduct the same evaluation on our PLG-MI results. These results, found in Tables 2 and 3 of the attached file Auther Rebuttal, provide a more comprehensive analysis of Trap-MID. We will add this analysis to our paper.
>
> **Q1: How sensitive is the Trap-MID to the hyper-parameters?**
>
> We discussed the impact of blend ratio in Appendix D.3 and trapdoor loss weight in Appendix D.4. According to Tables 7 and 8 in our paper, Trap-MID is not very sensitive to these hyper-parameters regarding test accuracy, maintaining about 81-84% accuracy under different configurations.
>
> For defense performance, Table 7 in our paper shows that a lower blend ratio generally provides better defense. There is an abrupt drop in attack accuracy against KED-MI and PLG-MI (e.g., PLG-MI's attack accuracy drops from 93.86% to 1.92% when the blend ratio decreases from 0.05 to 0.03), indicating that an invisible enough trigger is essential to mislead certain attacks.
>
> Additionally, Table 8 in our paper shows that increasing trapdoor loss weight from 0.02 to 0.2 reduces PLG-MI's attack accuracy from 23.84% to 6.23%.
>
> [1] Struppek et al., Plug & Play Attacks: Towards Robust and Flexible Model Inversion Attacks. ICML 2022
> [2] Wang et al., Variational model inversion attacks. NeurIPS 2021

---

> > ### Comment · Reviewer_BGpu · 2024-08-10
> > **Official Comment by Reviewer BGpu**
> >
> > I appreciate the authors' efforts in providing a thorough rebuttal and conducting additional experiments.
> >
> > Some of my concerns have been well addressed. However, a few issues remain:
> >
> > - The effectiveness of Trap-MID in defending against more practical MI setups (e.g., PPA attacks, high-resolution) is still limited.
> >
> > - My concerns regarding the presented visualization results still remains. While I appreciate the authors' effort in presenting additional results on other metrics and highlighting some examples, I noticed some inconsistencies between the visualization results (in the attached file and Appendix) and the quantitative results. For example, my observation (of course it is subjective) is that the visualizations suggest that Trap-MID and NegLS perform similarly in defending against PLG-MI (indeed Trap-MID could be slightly better). However, the quantitative results indicate that Trap-MID outperforms NegLS by over 80%.
> >
> > - I also agree with Reviewer P6SC that runtime is a critical aspect of Trap-MID that should be addressed to further strengthen the method.
> >
> > Besides, I have a few questions regarding the evaluation models. Could you confirm whether the exact evaluation models used in your experiments are the same as those from existing works, which are publicly available and widely used in MI research? Also, could you clarify the input resolution of these evaluation models for both high-resolution and low-resolution MI setups?

---

> > > ### Author Response · Authors · 2024-08-11
> > >
> > > Thank you for your thoughtful review and the time you've taken to provide valuable feedback. We address your remaining concerns below:
> > >
> > > **Q1: The effectiveness of Trap-MID in defending against more practical MI setups (e.g., PPA attacks, high-resolution) is still limited**
> > >
> > > While Trap-MID's defense performance against PPA may be limited with the default setup, it is worth noting that it can be effective with proper configuration. For instance, Table 2 in our attached file shows that Trap-MID reduces PPA's attack accuracy to 11.78% on DenseNet-169 when $\beta=0.5$. However, we acknowledge that tuning hyper-parameters is essential for optimal defense across different datasets and model architectures.
> > >
> > > **Q2: My observation is that the visualizations suggest that Trap-MID and NegLS perform similarly in defending against PLG-MI. However, the quantitative results indicate that Trap-MID outperforms NegLS by over 80%.**
> > >
> > > We understand your concern regarding the perceived inconsistencies between the visualizations and attack accuracy. We found that metrics based on the evaluation model, such as attack accuracy and KNN distance, estimate the extracted attributes, while FID better identifies unnatural or out-of-distribution outcomes. Successful attacks should perform well across all these metrics, recovering realistic images with private attributes.
> > >
> > > For instance, while MID, BiDO, and NegLS suffer from high attack accuracy (89-93%) against PLG-MI, NegLS's higher FID (69 vs. 14-17) suggests more unnatural recovered samples, indicating its better defense performance. Therefore, Trap-MID does not outperform these approaches to the same degree.
> > >
> > > Trap-MID’s lower attack accuracy and higher KNN distance imply that its recovered images reveal fewer private attributes than other methods (as discussed in our previous rebuttal). However, Trap-MID's slightly lower FID suggests that the recovered images from NegLS are slightly more unnatural (58 vs. 69), which might explain the similar visualization results observed between NegLS and Trap-MID.
> > >
> > > Further evaluations of PLG-MI's results are provided in Table 3 of our attached file:
> > >
> > > |Defense|Acc ↑|$\delta_{face}$ ↑|Precision ↓|Recall ↓|Density ↓|Coverage ↓|
> > > |-|-|-|-|-|-|-|
> > > |-|87.83|0.6110|19.39|13.17|0.0893|0.1498|
> > > |MID|76.67|0.6410|21.25|33.96|0.0913|0.1734|
> > > |BiDO|79.62|0.7058|20.17|10.17|0.0807|0.1367|
> > > |NegLS|81.76|0.7587|**3.80**|**0.00**|**0.0244**|**0.0189**|
> > > |Trap-MID|81.62|**1.3845**|9.56|71.63|0.0328|0.0728|
> > >
> > > $\delta_{face}$ measures the feature distance between recovered and private data using FaceNet pre-trained on VGGFace2, where the highest value for Trap-MID indicates fewer extracted facial attributes. In contrast, other metrics measured by InceptionV3 (same as FID) suggest the more unnatural outcomes produced from NegLS. Trap-MID ranks second to NegLS in most metrics, while its ability to classify arbitrary images with injected triggers as corresponding classes results in more diverse recovered images, leading to a broader manifold of recovered data and higher recall.
> > >
> > > **Q3: I also agree with Reviewer P6SC that runtime is a critical aspect of Trap-MID that should be addressed to further strengthen the method.**
> > >
> > > We agree that computational efficiency is an important aspect, despite the effectiveness of Trap-MID. We will include this consideration in our Future Work section.
> > >
> > > **Q4: Could you confirm whether the exact evaluation models used in your experiments are the same as those from existing works? Also, could you clarify the input resolution of these evaluation models for both high-resolution and low-resolution MI setups?**
> > >
> > > We apologize for the previous mistake regarding the input resolution in our rebuttal to Reviewer bcsy. The correct input size of the evaluation model in the low-resolution setting is 112x112, not 224x224 as previously mentioned. Below are the details about data resolution and the evaluation models used:
> > >
> > > - **Low-resolution setups (against GMI, KED-MI, LOMMA, PLG-MI, BREP-MI)**
> > >     - Target model input resolution: 64x64
> > >     - Adversary's generator output resolution: 64x64 (resized to fit target/evaluation models' input sizes)
> > >     - Evaluation model input resolution: 112x112
> > >     - We used the publicly available checkpoint provided in PLG-MI's official GitHub repository for the evaluation model, identical to that in previous works.
> > > - **High-resolution setups (against PPA)**
> > >     - Target model input resolution: 224x224
> > >     - Adversary's generator output resolution: 1024x1024 (center cropped to 800x800 and resized to fit target/evaluation models' input sizes)
> > >     - Evaluation model input resolution: 299x299
> > >     - Since the PPA's authors have not released their evaluation model checkpoint, we reproduced it using their official code and configuration file. While the models may not be identical, they were trained with the same setup. Our evaluation model's test accuracy is 91.34%, compared to 93.28% reported in their paper.

---

> > > > ### Comment · Reviewer_BGpu · 2024-08-12
> > > > **Official Comment by Reviewer BGpu**
> > > >
> > > > Thank you for your response and clarification.
> > > >
> > > > My concerns have been addressed to some extent.
> > > >
> > > > I note that the PPA authors have released their evaluation checkpoints in the releases section of their official GitHub. While the reproduced evaluation model you have used is somewhat acceptable, it isn't perfect. I suggest using the PPA's checkpoints with higher test accuracy for a more ideal evaluation and make the results consistent with other MI papers.
> > > >
> > > > Overall, I appreciate authors' thorough rebuttal. I will increase the score to 5 accordingly.
> > > >
> > > > Thank you.

---

> > > > > ### Author Response · Authors · 2024-08-12
> > > > >
> > > > > Thank you for pointing out the availability of PPA's evaluation checkpoints. We have re-evaluated our metrics using the official model, which generally indicates slightly lower attack accuracy than our initial analysis in Table 2 of the attached file. Despite differences in the scale of feature distances, both evaluation models show a consistent trend in defense performance.
> > > > >
> > > > > |Model|Defense|Acc ↑|AA-1 ↓|AA-5 ↓|$\delta_{eval}$ ↑|
> > > > > |-|-|-|-|-|-|
> > > > > |DenseNet-169|-|87.08|82.62|95.32|297.38|
> > > > > ||Trap-MID|89.98|58.12|71.90|355.56|
> > > > > ||$\beta$=0.5|83.39|9.18|17.40|504.63|
> > > > > ||w/ NegLS|83.56|**1.44**|**4.18**|**526.22**|
> > > > > |ResNet-152|-|87.08|84.10|96.40|312.19|
> > > > > ||Trap-MID|89.98|**72.40**|**86.62**|**333.12**|
> > > > > |ResNeSt-101|-|87.08|82.76|95.44|298.44|
> > > > > ||Trap-MID|89.98|**70.58**|**84.16**|**330.08**|
> > > > >
> > > > > We genuinely appreciate your constructive feedback and the effort you put into reviewing our work! Your insights are invaluable in improving our research.

---

### Official Review · Reviewer_P6SC · 2024-07-03

**Soundness:** 2
**Presentation:** 2
**Contribution:** 2
**Rating:** 4
**Confidence:** 4

**Summary:**

The paper proposes Trap-MID, a trapdoor-based defense mechanism to protect deep neural networks (DNNs) against Model Inversion (MI) attacks. The technique involves integrating trapdoors into the model to mislead MI attacks, causing them to extract trapdoor triggers rather than private data. The authors provide theoretical insights into the effectiveness and invisibility of these trapdoors and validate their approach through empirical experiments, demonstrating superior performance against various MI attacks without the need for extra data or significant computational overhead.

**Strengths:**

- Trap-MID presents the use of trapdoors to mislead MI attacks, filling a gap in the existing defense strategies.
- The empirical defense results seem good.

**Weaknesses:**

- The proposed method involves multiple optimization processes, making it computationally expensive and potentially impractical for large-scale or resource-constrained applications.
- According to Theorem 1, a more effective (larger $\delta$) and invisible (smaller $\epsilon$) trapdoor can lead to a larger lower bound to the expected posterior probability, making it more likely to be extracted by an MI attack. However, the paper does not analyze the discrepancy between the two posterior probability distributions, which is crucial for understanding the practical implications of the defense mechanism.
- The paper's organization and logical flow can be further improved for better readability and comprehension. The transitions between sections are sometimes abrupt, and a more cohesive structure would enhance the overall presentation.

**Questions:**

See weaknesses.

---

> ### Author Rebuttal · Authors · 2024-08-07
>
> Thank you for your valuable comments. We address the specific weaknesses raised in your review below:
>
> **W1: The proposed method involves multiple optimization processes, making it computationally expensive and potentially impractical for large-scale or resource-constrained applications.**
>
> We appreciate your concern. Here are the training times of different defense methods in our experiments:
>
> |Defense Method|Training Time|
> |-|-|
> | Unprotected model | 15 mins |
> | MID | 15 mins |
> | BiDO | 16 mins |
> | NegLS | 35 mins |
> | Trap-MID | 1 hour 15 mins |
>
> While Trap-MID does take the longest time due to three gradient updates per epoch (discriminator, triggers, and target model), it is worth noting that it also significantly surpasses other defenses against recent MI attacks. Moreover, Trap-MID still requires less data and computational cost than the existing trap-based defense, NetGuard, which demands an additional dataset, training an extra classifier, and conducting shadow attacks.
>
> We believe that it would be a valuable future direction to develop a more efficient trigger generation to make Trap-MID more practical for large-scale applications (e.g., pre-computing triggers with fewer steps to reduce overhead during model training).
>
> **W2: The paper does not analyze the discrepancy between the two posterior probability distributions, which is crucial for understanding the practical implications of the defense mechanism.**
>
> Theorem 1 establishes a lower bound for the posterior probability of poisoned data compared to benign data:
>
> $$\mathbb{E}\_{Y \sim p(Y)} \mathbb{E}\_{X \sim p(X)}[\log p_f(\Pi_y(X)|Y)] \ge \mathbb{E}\_{(X, Y) \sim p(X, Y)}[\log p_f(X|Y)] + (\delta - \epsilon)$$
>
> For example, since the unprotected model isn't injected with a trapdoor, it would have a negative trapdoor effectiveness $\delta$, leading to a lower expected posterior probability of poisoned data $p_f(\Pi_y(X)|y)$ than benign data $p_f(X|y)$ and making MI attacks more likely to extract private data.
>
> In contrast, a trapdoored model with a stronger predictive power on invisibly triggered data (especially when $\delta > \epsilon$) would result in a higher expected posterior probability for poisoned data compared to benign data, misleading MI attacks to recover triggered data instead.
>
> **W3: The paper's organization and logical flow can be further improved for better readability and comprehension.**
>
> Thank you for highlighting this issue. We will refine our explanations and improve the organization for better readability. We would greatly appreciate any specific feedback on the sections that need improvement to help us enhance our presentation and address any confusion.

---

> > ### Comment · Reviewer_P6SC · 2024-08-12
> >
> > Thank you for your effort in providing a thorough response, I decided to slightly increase my score. After reading all the reviews and the corresponding rebuttals, I believe that the paper still has significant room for improvement, and I encourage the authors to take all the comments into account to further refine and enhance the manuscript.

---

> > > ### Author Response · Authors · 2024-08-12
> > >
> > > Thank you for your thoughtful feedback! We will carefully review all the comments and suggestions to further refine and enhance our paper.

---

### Official Review · Reviewer_MuQ4 · 2024-07-11

**Soundness:** 3
**Presentation:** 3
**Contribution:** 2
**Rating:** 5
**Confidence:** 3

**Summary:**

This paper presents Trap-MID, a novel defense mechanism against model inversion attacks that utilizes trapdoor injection techniques. By incorporating a trapdoor into the model, Trap-MID misleads MI attacks into extracting trapdoor information instead of private data, effectively preserving privacy. The paper contributes to the field by:
1. Introducing a Novel Defense Mechanism: Trap-MID pioneers the exploration of the relationship between trapdoor injection and MI defense, providing a new approach to tackle this challenging privacy problem.
2. Theoretical Insights and Empirical Validation: The paper provides theoretical analysis on the impact of trapdoor effectiveness and visibility on deceiving MI attacks and validates its effectiveness through extensive experiments, showcasing its superior performance compared to existing defenses.

**Strengths:**

+ New Application of Trapdoors.The paper creatively applies trapdoor injection techniques, traditionally used for adversarial detection, to defend against model inversion attacks. This novel application opens up new avenues for privacy-preserving DNNs.
+ Theoretical Analysis: The paper presents a solid theoretical foundation for trapdoor-based defenses, establishing definitions for trapdoor effectiveness and visibility and providing a theorem that explains their impact on MI attacks.
+ Comprehensive Experiments: The authors conduct thorough experiments on various MI attacks using different DNN architectures and datasets, demonstrating the generalizability and robustness of Trap-MID.
+ Well-structured and Clear Explanation: The paper is well-structured and provides clear explanations of the proposed method, theoretical analysis, and experimental results. The figures and tables effectively illustrate the key concepts and findings.

**Weaknesses:**

-  Assumption of Trust: Trap-MID assumes a level of trust between data providers and the model owner, which might not always be feasible in practice. Future work could explore ways to empower data providers or individuals to secure their sensitive information before sharing data.
- Limited Scope of Attackers: The paper assumes white-box attackers with full access to the model. However, in practical scenarios, attackers may only have access to the model's predictions or even just the labels. It would be beneficial to explore the effectiveness of Trap-MID against black-box and label-only attackers.
- Vulnerability to KD: Trap-MID's efficacy may be limited against MI attacks involving knowledge distillation (KD), as KD enables the attacker to extract trapdoor information and explore private data simultaneously. Developing more robust trapdoors against KD is a crucial direction for future research.
- Robustness to Adaptive Attacks: The paper explores the effectiveness of Trap-MID against adaptive attacks where the attacker knows the existence of trapdoors. However, it would be valuable to investigate the robustness of Trap-MID against other types of adaptive attacks, such as those that exploit specific vulnerabilities in the trapdoor design.
- Random Trigger Initialization: The randomly initialized triggers introduce variability in defense performance, leading to a larger standard deviation compared to previous defenses. Exploring more stable and powerful trigger designs would enhance the consistency and reliability of Trap-MID.
- Limited Exploration of Trigger Design: While the paper demonstrates the effectiveness of a simple trigger design, exploring more sophisticated trigger designs could potentially improve the defense performance further.
- Limited Analysis of Other Reconstruction-based Attacks: The paper primarily focuses on model inversion attacks. Analyzing the effectiveness of Trap-MID against other reconstruction-based attacks, such as Gradient Inversion Attacks and Embedding Inversion Attacks, would provide a more comprehensive evaluation of its generalizability.

**Questions:**

1. Impact of KD on Defense Performance: Could the authors provide more insights into the impact of KD on Trap-MID's defense performance? Specifically, what modifications could be made to the trapdoor design or training process to make it more robust against KD-based attacks?
2. Stability of Trigger Design: How sensitive is Trap-MID's performance to the specific trigger design? Are there any guidelines or heuristics for choosing an effective and stable trigger design?
3. Trade-off between Privacy and Utility: The paper mentions an accuracy-privacy trade-off due to the trapdoor loss weight. Could the authors elaborate on this trade-off and provide more insights into the impact of different weight values on privacy and utility?
4. Impact of Distributional Shifts: The paper briefly mentions distributional shifts in the auxiliary dataset. Could the authors provide a more detailed analysis of how distributional shifts impact Trap-MID's defense performance and how it compares to existing defenses in such scenarios?
5. Comparison with Other Reconstruction-based Attacks: The paper focuses on model inversion attacks. How effective is Trap-MID against other reconstruction-based attacks, such as Gradient Inversion Attacks and Embedding Inversion Attacks?

**Limitations:**

The authors acknowledge the limitations of their work, including the assumption of trust and vulnerability to KD. They also recognize the need for further exploration of trigger designs and analysis of other reconstruction-based attacks. These limitations are clearly stated

---

> ### Author Rebuttal · Authors · 2024-08-07
>
> Thank you for your detailed and insightful review. We address the specific weaknesses and questions raised in your comments below:
>
> **W1: Trap-MID assumes trust between data providers and model owners, which isn't always feasible.**
>
> This limitation is discussed in Appendix A.2. We follow a common scenario that relies on model owners to protect private data. Since backdoor attacks can be conducted by poisoning the dataset, we believe Trap-MID's success can inspire future work enabling data owners to preserve privacy.
>
> **W2: It would be beneficial to explore the effectiveness of Trap-MID against black-box and label-only attacks.**
>
> To address this, we conduct experiments against BREP-MI, a label-only attack. Our result shows that BREP-MI is ineffective on Trap-MID, requiring over 820,000 iterations to initialize latents for all identities and getting 0% attack accuracy in untargeted attacks (recovering 300 identities), demonstrating Trap-MID's efficacy and generalizability:
>
> |Defense|Accuracy ↑|# of Initial Iterations ↑|Attack Accuracy ↓|
> |-|-|-|-|
> |Trap-MID|81.62|171|0.00|
>
> Results against other defenses can be found in Table 1 of the attached file in Author Rebuttal. We will add this analysis to our paper.
>
> **W3 & Q1: Could the authors provide more insights into KD's impact on Trap-MID and how to make it more robust against KD-based attacks?**
>
> KD reduces Trap-MID's efficacy as student models typically do not learn trapdoor behaviors. Conducting shadow KD during trapdoor injection could be one of the solutions [1]. Besides, we found that combining Trap-MID with NegLS also improves defense, even against KD-based attacks:
>
> |Attack|Defense|Accuracy ↑|AA-1 ↓|AA-5 ↓|KNN Dist ↑|FID ↑|
> |-|-|-|-|-|-|-|
> |LOMMA (KED-MI)|Trap-MID|81.37|61.25|85.76|1404.77|24.19|
> ||w/ NegLS|77.10|**42.47**|**70.64**|**1521.82**|**37.22**|
>
> Intuitively, NegLS makes it harder to extract private data and therefore makes trapdoor more attractive. This suggests Trap-MID to be an orthogonal work to existing defenses, and a hybrid approach may enhance robustness against specific adaptive attacks.
>
> Results against other attacks are available in Table 4 of the attached file. We will add this analysis to our paper.
>
> **W4: It would be valuable to investigate Trap-MID's robustness against other types of adaptive attacks.**
>
> Thank you for the suggestions. Due to time constraints, we couldn't explore this further in this work. We recognize the importance of this future direction to help improve the trapdoor design.
>
> **W5: The randomly initialized triggers introduce variability in defense performance.**
>
> We acknowledged this limitation in Appendix A.2. However, the worst-case performance of Trap-MID still surpasses the best-case performance of existing defense against most attacks, indicating its effectiveness:
>
> |Attack|Defense|Acc ↑|AA-1 ↓|AA-5 ↓|KNN Dist ↑|FID ↑|
> |-|-|-|-|-|-|-|
> |GMI|BiDO (best)|78.32|4.42|12.94|2036.78|47.55|
> ||Trap-MID (worst)|79.39|**0.56**|**2.46**|**2280.19**|**75.16**|
> |KED-MI|NegLS (best)|81.79|29.64|57.28|1544.90|**47.31**|
> ||Trap-MID (worst)|81.55|**23.80**|**46.58**|**1665.86**|21.23|
> |LOMMA (GMI)|NegLS (best)|81.79|48.58|75.80|1423.78|**38.27**|
> ||Trap-MID (worst)|81.55|**44.80**|**72.60**|**1535.01**|35.47|
> |LOMMA (KED-MI)|MID (best)|76.67|**59.18**|**86.30**|**1413.53**|**24.55**|
> ||Trap-MID (worst)|81.55|69.32|90.90|1333.36|20.95|
> |PLG-MI|NegLS (best)|81.79|83.84|96.58|1495.23|**73.45**|
> ||Trap-MID (worst)|79.39|**15.72**|**30.98**|**1843.42**|36.91|
>
> **W6: Exploring more sophisticated trigger designs could improve the defense further.**
>
> This future direction is discussed in Appendix A.2. Our work serves as the first step in trapdoor-based defense. Besides, Appendix D.1 compares Trap-MID with patch-based triggers, showing the importance of trigger design. Future research could explore advanced triggers to improve defense further.
>
> **W7 & Q5: How effective is Trap-MID against other reconstruction-based attacks?**
>
> We discuss this future work in Appendix A.2. As these reconstruction attacks also optimize inputs to satisfy the adversary's objectives, we believe that Trap-MID can be extended to mitigate them with proper "shortcuts." However, since this work focuses on MI defense, the extension is beyond the scope, and we leave it to future research.
>
> **Q2: How sensitive is Trap-MID to the specific trigger design? Are there any guidelines for choosing a stable trigger design?**
>
> Appendix D.1 compares Trap-MID with patch-based triggers, showing the importance of trigger design. In this work, we use trapdoor loss and discriminator loss to optimize the triggers, which already surpassed existing defenses. We leave further customization to future work (e.g., improving trigger initialization and optimization, or using recent backdoor techniques).
>
> **Q3: Could the authors elaborate on the accuracy-privacy trade-off from trapdoor loss weight?**
>
> We discussed this trade-off in Appendix D.4. A larger weight encourages the model to learn trapdoor behavior instead of the main task, enhancing defense at the cost of accuracy. Table 8 shows that increasing this weight from 0.02 to 0.2 reduces PLG-MI's attack accuracy from 24% to 6%, while the test accuracy decreases from 84% to 81%.
>
> **Q4: Could the authors provide a more detailed analysis of how distributional shifts impact Trap-MID's defense performance and how it compares to existing defenses in such scenarios?**
>
> Section 4.2 and Appendix E.2 analyze this impact. Distributional shifts in auxiliary data degrade attack performance, especially against Trap-MID. In particular, distributional shifts make extracting private data harder, which therefore makes trapdoors more attractive to the attacks. Tables 4 and 11 show that PLG-MI's attack accuracy drops from 6% to nearly 0% with distributional shifts, highlighting its effectiveness in this case.
>
> [1] Ge et al., Anti-Distillation Backdoor Attacks: Backdoors Can Really Survive in Knowledge Distillation, ACM MM 2021

---

> > ### Comment · Reviewer_MuQ4 · 2024-08-10
> >
> > Thanks for the author's reply. The rebuttal basically solved my questions, so I will slightly increase the score based on the original rating.

---

> > > ### Author Response · Authors · 2024-08-10
> > >
> > > Thank you for your reconsideration! We are glad to hear that our responses have successfully addressed your concerns.

---

> > > > ### Author Response · Authors · 2024-08-13
> > > >
> > > > Dear Reviewer,
> > > >
> > > > Thank you once again for taking the time to review our manuscript and for your valuable feedback. Based on your previous comment about slightly increasing the score, we would like to kindly remind you to update your score to reflect your final rating accurately. We sincerely appreciate your efforts in helping us improve our work.

---

### Author Rebuttal · Authors · 2024-08-07

We thank the reviewers for their valuable feedback that helped us improve our paper. We are encouraged that they found Trap-MID to be a **novel** (MuQ4, bcsy), **clever, and conceptually straightforward** (bcsy) approach to defending against MI attacks. The feedback that our work **fills a gap in existing defenses** (P6SC), **provides new insights** (BGpu), and **opens up avenues for privacy-preserving DNNs** (MuQ4) is greatly appreciated. We are pleased that the experimental results (bcsy), defense performance (MuQ4, P6SC, bcsy), and efficiency of Trap-MID were well-received (P6SC). We are also glad that our paper was found to be **well-structured** (MuQ4) and **well-written** (bcsy), with an **intuitive idea** (BGpu), **solid theoretical foundation** (MuQ4), **comprehensive experiments** (MuQ4), and **well-defined evaluation** (bcsy).

We address some common points below. Detailed responses to other questions are in the reviewer-specific rebuttals. Additional tables and figures referenced in our responses are in the attached PDF.

**Q1: Defense performance against black-box/label-only attacks (MuQ4, bcsy)**

We add the experiments against BREP-MI, a label-only attack. BREP-MI failed to initialize latents for all 1,000 identities in a reasonable time against Trap-MID, requiring over 820,000 iterations to sample latents for only 942 of them. This demonstrates Trap-MID's effective privacy protection.

We also conducted untargeted attacks to recover 300 identities. Trap-MID significantly increased the number of initial iterations required and reduced attack accuracy to 0%:

|Defense|Accuracy ↑|# of Initial Iterations ↑|Attack Accuracy ↓|
|-|-|-|-|
|-|87.83|2|65.00|
|MID|76.67|2|46.33|
|BiDO|79.62|3|39.00|
|NegLS|81.76|3|52.00|
|Trap-MID|81.62|**171**|**0.00**|

The results can also be found in Table 1 of the attached file. We will add this analysis to our paper.

**Q2: Defense performance under high-resolution scenarios with modern architectures (BGpu, bcsy)**

We add the experiments against PPA to demonstrate a high-resolution (224x224) scenario with modern models. Due to time and computational constraints, we modified the attack to optimize 20 samples and select 5 in the final stage (PPA originally optimized 200 samples and selected 50, which would take about 10 days on our machine).

Although Trap-MID does not fully mitigate PPA with default settings, it preserves privacy to certain degrees without an accuracy drop. Besides, increasing trapdoor loss weight or combining Trap-MID with NegLS further improves defense performance, reducing attack accuracy to 2%, which shows Trap-MID's effectiveness under different scenarios. We discuss more about the defense combination in Q4.

|Defense|Acc ↑|AA-1 ↓|AA-5 ↓|$\delta_{face}$ ↑|$\delta_{eval}$ ↑|FID ↑|
|-|-|-|-|-|-|-|
|-|87.08|92.00|98.28|0.7240|140.93|37.76|
|Trap-MID|89.98|59.48|72.22|0.9468|182.90|49.64|
|$\beta$=0.5|83.39|11.78|19.24|1.3985|269.99|63.04|
|w/ NegLS|83.56|**1.60**|**4.90**|**1.4744**|**279.68**|**79.37**|

Detailed results, including other architectures, are available in Table 2 of the attached file. We will add this analysis to our paper.

**Q3: Evaluation metrics (BGpu, bcsy)**

We agree that the visualization can be subjective due to the ambiguous definition of "class representative." However, in Figure 2 of our paper, we found that recovered images from Trap-MID show more different attributes from private data. For example, those of Identity 1 show different skin tones, those of Identity 2 and 5 show different hairstyles, and those of Identity 4 show different genders. We also provide more recovered samples in Figure 1 of the attached file. Similarly, the recovered images of Identity 4 from Trap-MID show different hair colors from private data compared with other defenses.

For the quantitative analysis, we found that attack accuracy/KNN distance and FID can complement each other. Specifically, the formers estimate the extracted attributes of target identity, while the latter measures the naturalness and styling similarity of recovered images. However, we agree that developing a universal metric capturing all these properties would be valuable for straightforward comparisons between MI attacks/defenses.

Our PPA experiments in Q2 include additional metrics such as FaceNet's feature distance used in [1], as well as improved precision, recall, density, and coverage used in [2]. Also, we use them to evaluate PLG-MI's results to provide a more comprehensive analysis of Trap-MID. The results are available in Tables 2 and 3 of the attached file. We will add this analysis to our paper.

**Q4: Further design of adaptive attacks. How to make Trap-MID more robust to mitigate them? (MuQ4, bcsy)**

We acknowledge the need for exploring adaptive attacks further. Our work is a first step in understanding the relationship between trapdoors and MI defenses. Future research could develop advanced adaptive attacks to investigate and improve the trapdoor design further.

In addition, we found that combining Trap-MID and NegLS improves defense performance, even against KD-based attacks, suggesting Trap-MID to be an orthogonal work to existing defenses and can be incorporated with them. Intuitively, previous approaches focus on reducing information leakage, which makes it harder to extract private data, and therefore makes our shortcut more attractive. Another future direction is to combine multiple defense methods to counter specific adaptive attacks.

|Attack|Defense|Accuracy ↑|AA-1 ↓|AA-5 ↓|KNN Dist ↑|FID ↑|
|-|-|-|-|-|-|-|
|LOMMA (KED-MI)|Trap-MID|81.37|61.25|85.76|1404.77|24.19|
||w/ NegLS|77.10|**42.47**|**70.64**|**1521.82**|**37.22**|

Results against other attacks can be found in Table 4 of the attached file. We will add this discussion and analysis to our paper.

[1] Struppek et al., Plug & Play Attacks: Towards Robust and Flexible Model Inversion Attacks. ICML 2022
[2] Wang et al., Variational model inversion attacks. NeurIPS 2021

---

### Decision · Program_Chairs · 2024-09-25

**Decision:**

Accept (poster)

**Comment:**

The rebuttal addressed some of the concerns raised by the reviewers. However, there are still some concerns that need to be addressed to improve the paper. I believe the submission has merit and can be accepted conditioned on incorporating the discussion from rebuttal to the final draft. Also, addressing remaining concerns can definitely improve the quality of the paper.